# ByteFlow: Language Modeling through Adaptive Byte Compression without a Tokenizer

**Chunyuan Deng**[*]
Rice University
cd110@rice.edu

**Sanket Lokegaonkar**
Amazon Science
sllokega@amazon.com

**Colin Lockard**
Amazon Science
clockard@amazon.com

**Besnik Fetahu**
Amazon Science
besnikf@amazon.com

**Nasser Zalmout**
Amazon Science
nzalmout@amazon.com

**Xian Li**
Amazon Science
xianlee@amazon.com

## Abstract

Modern language models (LMs) still rely on fixed, pre-defined subword tokenizations. Once a tokenizer is trained, the LM can only operate at this fixed level of granularity, which often leads to brittle and counterintuitive behaviors even in otherwise strong reasoning models. We introduce **ByteFlow Net**, a new hierarchical architecture that removes tokenizers entirely and instead enables models to learn their own segmentation of raw byte streams into semantically meaningful units. ByteFlow Net performs compression-driven segmentation based on the coding rate of latent representations, yielding adaptive boundaries *while preserving a static computation graph via Top-K selection*. Unlike prior self-tokenizing methods that depend on brittle heuristics with human-designed inductive biases, ByteFlow Net adapts its internal representation granularity to the input itself. Experiments demonstrate that this compression-based chunking strategy yields substantial performance gains, with ByteFlow Net outperforming both BPE-based Transformers and previous byte-level architectures. These results suggest that end-to-end, tokenizer-free modeling is not only feasible but also more effective, opening a path toward more adaptive, robust, and information-grounded LMs.

## 1 Introduction

Tokenization is a foundational step in every language model pipeline (Grattafiori et al., 2024; Team et al., 2025; DeepSeek-AI et al., 2025; Yang et al., 2025). The model's first action is to segment raw input—be it text, code, or other modalities—into discrete tokens. This seemingly simple decision carries profound consequences, defining the model's vocabulary, sequence lengths, and the very granularity of its learned representations. The primary limitation of dominant strategies like byte-pair encoding (BPE) (Sennrich et al., 2015; Gallé, 2019; Liu et al., 2025) is their *static* nature. After training, they apply a fixed segmentation logic to all inputs, ignoring context, linguistic nuance, or task-specific requirements. This *static property on subword level* is the source of many wierd model behaviors, such as difficulties with counting, arithmetic, structured data, and multilingual text (Rust et al., 2020; Zhang et al., 2024; Yehudai et al., 2024). At a more fundamental level, tokenization introduces a non-learnable stage into the pipeline, breaking the end-to-end language modeling. This imposes a rigid inductive bias, forcing the model to expend its FLOPs on predefined units rather than learning how to allocate them dynamically.

Recent efforts to eliminate tokenizers have largely converged on hierarchical architectures. The central challenge for such designs is defining the high-level semantic units beyond byte level. Current methods generally fall into two main categories: i) *Heuristic-based* strategies that employ static chunking via fixed strides, word boundaries, or regular expressions (Yu et al., 2023b; Slagle, 2024a; Videau et al., 2025), and ii) *Dynamic chunking* that learn to segment sequences using a neural network, entropy thresholds, or cosine similarity (Nawrot et al., 2022; Pagnoni et al., 2025; Hwang

---

[*]Work primarily done during internship at Amazon Pretraining Team.

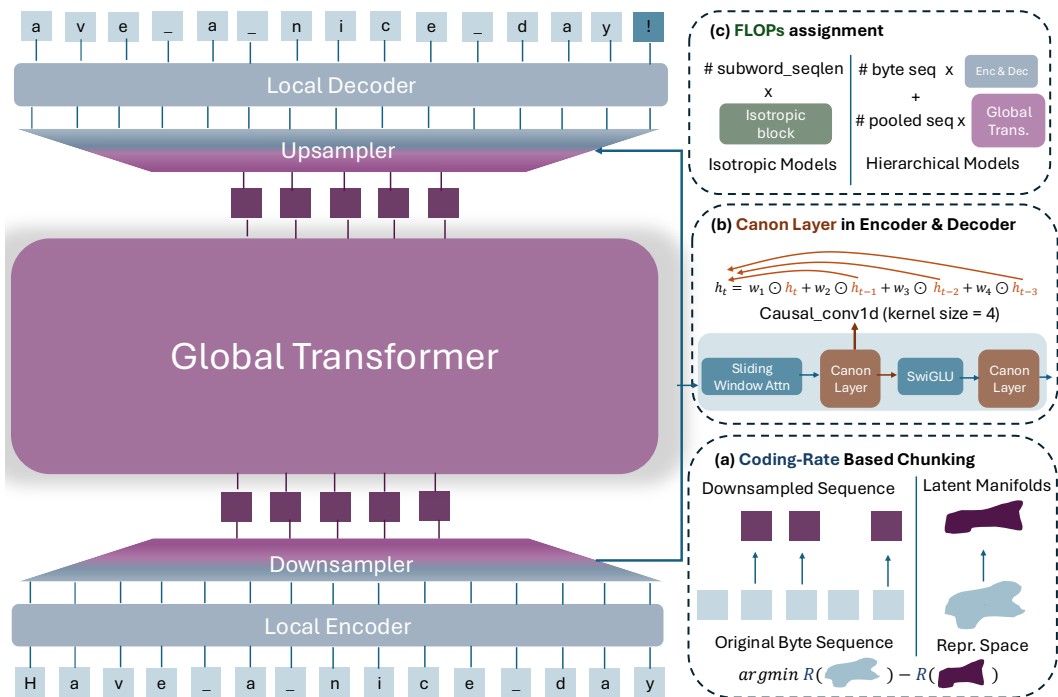

Figure 1: **Architecture of ByteFlow Net**. **(a)** ByteFlow Net's chunking strategy is primarily driven by the coding rate $R$ of latent representations. As shown in the figure, the model is encouraged to select token boundaries that form pooled subsequences which best compress the original input. **(b)** Since byte-level sequences are roughly $4\times$ longer, directly applying $O(n^2d)$ softmax attention becomes prohibitively expensive. To address this, we adopt sliding-window attention (SWA) combined with canon layers (Allen-Zhu, 2025), enabling efficient and low-cost token mixing. **(c)** The beauty of the hierarchical architecture lies in allocating the majority of FLOPs operating at the high-level information (a deep and wide global transformer), while using lightweight local encoders/decoders (shallow and narrow) to quickly process low-level information.

et al., 2025b)[1]. While heuristic approaches embed strong inductive biases into the model, dynamic methods introduce considerable uncertainty into the chunking process, which can hinder pattern finding during the early stages of pre-training. Furthermore, we still lack a dynamic mechanism for guiding the model's allocation of FLOPs in a principled manner.

We introduce *ByteFlow Net* (Figure 1), a novel hierarchical byte-level architecture that learns to *self-tokenize* directly from raw byte streams. Rather than applying a fixed vocabulary, ByteFlow Net integrates segmentation into its forward computation: as bytes flow through the network, it dynamically promotes them to higher-level calculations. The decision of when to commit a boundary is framed as a principled, *coding-rate-based compression* problem, estimating the representational cost of promoting the position to a higher level. This formulation turns boundary placement into an *online* information-theoretic optimization problem, enabling the model to adjust token granularity according to input complexity on its own.

Architecturally, ByteFlow Net follows a simple but effective hierarchy: The process begins with a *local encoder* that transforms byte sequences into contextualized representations. Next, a *chunking module* applies the coding-rate criterion to these representations, producing higher-level tokens on the fly. These dynamic tokens are then modeled by a *global transformer* to capture the deep and abstract patterns in high resolution level, before a decoder maps the global context back to byte-level predictions. Because this entire boundary selection process is integrated into the model's computation, ByteFlow Net naturally adapts across languages and domains without requiring language-specific rules or a separate tokenizer training stage.By unifying segmentation and representation learning, ByteFlow Net ensures that the boundaries of higher-level tokens are optimized to maximize information density for the global transformer. This transition from a fixed, discrete preprocessing

---

[1]H-Net (Hwang et al., 2025a) is our concurrent work that also explores end-to-end tokenizer-free modeling. We contrast our chunking approach with theirs in § 4.4.

step to a continuous, learnable bottleneck allows the model to preserve nuances—such as morphological variations or structural symbols—that are often lost or fragmented by static tokenizers.

**Contributions.**

- We introduce a new paradigm that replaces static tokenization with dynamic, learned segmentation. Our architecture, ByteFlow Net, operates end-to-end on raw bytes, using a principled information-theoretic objective to identify meaningful units on the fly.
- We demonstrate superior performance and scaling through extensive experiments. ByteFlow Net consistently outperforms both strong LLaMA baseline and other byte-level architectures on pre-training loss and downstream tasks, showing that end-to-end, tokenizer-free modeling is not only feasible but more effective.
- We reveal that the success of our approach stems from *its ability to preserve a coherent latent manifold*. Our ablation studies show that the coding-rate objective uniquely maintains the geometric structure of the data's representation, preventing the fragmentation that plagues other methods and enabling more powerful learning.

## 2 RELATED WORK

**Tokenizer-free Architecture.** Modern tokenizer-free architectures can be broadly categorized into three main approaches:

- **Pure Byte-Level Modeling:** These models perform language modeling directly on raw byte sequences (Xue et al., 2022a). Given that the $O(n^2d)$ complexity of full attention is prohibitive for long sequences, architectures like MambaByte (Wang et al., 2024) have emerged as an effective solution, balancing fine-grained information processing with computational efficiency.

- **Hierarchical Modeling with Heuristic Chunking:** These methods use fixed, rule-based strategies to group bytes into larger units. For instance, MegaByte (Yu et al., 2023b) uses a fixed stride (e.g., 4 or 6 bytes) to create a higher level of representation, outperforming pure byte-level models while significantly saving FLOPs. Building on this, SpaceByte (Slagle, 2024b) uses word boundaries for chunking, achieving performance on par with or even exceeding BPE-based transformers on some pre-training corpora. AU-Net (Videau et al., 2025) further refines this concept by replacing simple word boundaries with a flexible set of regex rules to better handle special tokens and digits.

- **Hierarchical Modeling with Dynamic Chunking:** Instead of fixed rules, these models employ a learned mechanism to determine chunk boundaries. Nawrot et al. (2022; 2023); Kallini et al. use a neural network to gate token boundaries. The Byte Latent Transformer (BLT) (Pagnoni et al., 2024) first trains a separate entropy model and then uses it as a proxy to set a global chunking threshold. This multi-stage process is not fully end-to-end and functions more like a different tokenizer. In a concurrent work, H-Net (Hwang et al., 2025a) uses the cosine similarity between neighboring representations to decide chunking.

**Tokenization in Language Modeling.** The prevailing solution in modern LMs is subword tokenization (Sennrich et al., 2015; Kudo & Richardson, 2018; Zouhar et al., 2023; Schmidt et al., 2024; Liu et al., 2025) (e.g., BPE), which use a fixed-size vocabulary of word pieces to represent any text. These fixed vocabularies create a rigid, non-learnable stage in the modeling pipeline, often causing brittle and unexpected behaviors (Belinkov & Bisk, 2018; Sun et al., 2020; Rust et al., 2020; Petrov et al., 2023; Schmidt et al., 2024; Zhang et al., 2024; Yehudai et al., 2024), which motivated the development of modern tokenizer-free models that operate directly on raw bytes.

## 3 BYTEFLOW NET

**Overview.** ByteFlow Net is a hierarchical architecture that operates through five main stages: local encoder, downsampling (coding-rate chunking), global modeling, upsampling, and decoder:

$$x_{1:T} \in V^T \xrightarrow{\text{Local Encoder}} h_{1:T} \in \mathbb{R}^{T \times d_{\text{local}}} \quad \text{(contextualized byte representations)} \tag{1}$$

$$\xrightarrow{\text{Downsampling}} z_{1:K} \in \mathbb{R}^{K \times d_{\text{global}}} \quad \text{(adaptive chunking, } K \ll T) \tag{2}$$

$$\xrightarrow{\text{Global Transformer}} g_{1:K} \in \mathbb{R}^{K \times d_{\text{global}}} \quad \text{(High-resolution level modeling)} \tag{3}$$

$$\xrightarrow{\text{Upsampling}} s_{1:T} \in \mathbb{R}^{T \times d_{\text{local}}} \quad \text{(reconstruct to original length)} \tag{4}$$

$$\xrightarrow{\text{Decoder}} \hat{p}(x_{t+1}|\cdot) \in V \quad \text{(next byte prediction)} \tag{5}$$

Here $T$ is the input sequence length, $V \in \Delta^{258}$ (contains 256 UTF-8 Byte plus two *BOS/EOS* tokens) is the byte vocabulary, and $d_{\text{local}}, d_{\text{global}}$ are the hidden dimensions at local and global levels.

## 3.1 LOCAL ENCODER: FAST PROCESSING OVER BYTE-LEVEL REPRESENTATIONS

The local encoder are stacked small transformer. The input byte sequence $x_{1:T} \in V^T$ first embedded into a continuous representation $h_{1:T}^{(0)}$ by the learned byte embedding matrix, then transformed into contextualized representations $h_{1:T} \in \mathbb{R}^{T \times d_{\text{local}}}$.

**Transformer Blocks with Sliding Window Attention.** We stack $E$ pre-norm causal transformer blocks. For each layer $l \in \{1, \ldots, E\}$ and position $t \in \{1, \ldots, T\}$:

$$u_t^{(l)} = \text{LN}\big(h_t^{(l-1)}\big), \tag{6}$$

$$\hat{h}_t^{(l)} = Canon(h_t^{(l-1)} + \text{SWA}(\mathbf{Q}, \mathbf{K}, \mathbf{V})), \mathbf{Q}, \mathbf{K}, \mathbf{V} = u_{1:t}^{(l)} X_{\mathbf{Q}}, u_{1:t}^{(l)} X_{\mathbf{K}}, u_{1:t}^{(l)} X_{\mathbf{V}} \tag{7}$$

$$v_t^{(l)} = \text{LN}\big(\hat{h}_t^{(l)}\big), \tag{8}$$

$$h_t^{(l)} = Canon(\hat{h}_t^{(l)} + \text{SwiGLU}\big(v_t^{(l)}\big)), \tag{9}$$

where: $\text{LN}(\cdot)$ denotes layer normalization. $\text{SWA}(\cdot)$ represents sliding window attention (SWA) with window size $w_{\text{local}}$. This reduces computational complexity from $O(T^2)$ to $O(T \cdot w_{\text{local}})$.

$\text{SwiGLU}(\cdot)$ is the gated activation function $\text{SwiGLU}(x) = \text{Swish}(xW_1) \odot (xW_2)$, where $W_1, W_2 \in \mathbb{R}^{d_{\text{local}} \times d_{\text{ff}}}$ are learned projection matrices, $d_{\text{ff}}$ is the feed-forward hidden dimension, $\text{Swish}(x) = x \cdot \sigma(x)$ with $\sigma(\cdot)$ being the sigmoid function, and $\odot$ denotes element-wise multiplication.

**Canon Layer.** Canon layer are introduced in Allen-Zhu (2025) to foster the token mixing:

$$Canon(h_t) = w_0 \odot h_t^{(l)} + w_1 \odot h_{t-1}^{(l)} + w_2 \odot h_{t-2}^{(l)} + w_3 \odot h_{t-3}^{(l)}, \tag{10}$$

where $w_i \in \mathbb{R}^{d_{\text{local}}}$ are learned gating vectors. They are basically `causal_conv1d` with kernel size = 4, so highly efficient CUDA operator are supported.

**Why SWA + Canon Layer for Token Mixing.** Theoretically if we use SWA along, given a sequence length $T$ and window size $w_{\text{local}}$, we will need at least $\frac{T}{w_{\text{local}}}$ encoder layers to ensure every byte position can attend to every other. This would necessitate a very deep local encoder for long sequences, increasing computational cost and potentially hindering training stability. The canon layer instead is an efficient addition, as it introduces negligible parameter overhead and benefits from highly optimized implementations.

## 3.2 DOWNSAMPLING: CODING-RATE CHUNKING

The chunker then determines which byte positions to promote to the next hierarchical level by evaluating the *coding rate* of contextualized representations. This approach is grounded in information theory: positions with high coding rates contain more information and should be preserved as chunk boundaries, while positions with low coding rates can be safely compressed away.

**Lossy Coding Rate in Representation Space.** Let the local encoder produce contextualized representations $h_{1:T} \in \mathbb{R}^{T \times d_{\text{local}}}$. The lossy coding rate (Cover, 1999; Ma et al., 2007)[2] for

---

[2]We provide theoretical derivation in Appendix A and a fast approximation in Appendix B.

$h_{1:T} \in \mathbb{R}^{T \times d_{\text{local}}}$ is:

$$R_\varepsilon(h_{1:T}) = \frac{1}{2} \log \det \left( I + \frac{d_{\text{local}}}{\varepsilon^2} h_{1:T} h_{1:T}^\top \right), \tag{11}$$

where $\varepsilon^2$ is a noise variance parameter that controls the sensitivity of the coding rate computation.

$R_\varepsilon(h_{1:T})$ is large when the representation $h_{1:T}$ has large eigenvalues and spans diverse directions in the representation space, indicating high information position that warrants preservation.

**Streaming Decision.** Let the local encoder produce contextualized representations $h_{1:T} \in \mathbb{R}^{T \times d_{\text{local}}}$. The marginal coding rate at position $t$ measures the information gain from including the $t$-th byte:

$$\Delta R_t = R_\varepsilon(h_{1:t}) - R_\varepsilon(h_{1:t-1}). \tag{12}$$

$\Delta R_t$ is large when position $t$ introduces large information gain, indicating a natural segmentation boundary. Given the target global sequence length $K$, the chunking procedure begins by computing marginal coding rates $\Delta R_t$ for all positions $t \in \{2, 3, \ldots, T\}$. We initialize the selected positions with $\mathcal{S} = \{1\}$ to always include the BOS token, then identify the $(K - 1)$ positions with the largest $\Delta R_t$ values. Finally, we sort these selected positions chronologically to obtain $\mathcal{S} = \{s_1, s_2, \ldots, s_K\}$ where $s_1 = 1$ and $s_1 < s_2 < \cdots < s_K$. During teacher-forced training, Top-$K$ uses the full-sequence *importance* profile to allocate global compute, but causal masks ensure predictions never access future byte content.

After selecting K positions[3], we extract the corresponding representations $[h_{s_1}, h_{s_2}, \ldots, h_{s_K}] \in \mathbb{R}^{K \times d_{\text{local}}}$ and map them to the global representation space: $z_{1:K} = [h_{s_1}, h_{s_2}, \ldots, h_{s_K}]W_{\text{proj}} \in \mathbb{R}^{K \times d_{\text{global}}}$, where $W_{\text{proj}} \in \mathbb{R}^{d_{\text{local}} \times d_{\text{global}}}$ is the projection matrix.

**Why Not Global Threshold?** Instead of using a global information threshold for chunking, we select the Top-K positions with the highest information gain for two reasons. First, determining an appropriate global threshold is *non-trivial*: it often requires extensive empirical tuning and results in a "magic number" that is difficult to interpret or generalize. Second, a fixed threshold leads to dynamic chunks for different inputs. This variability in the global sequence length breaks the static computation graph. While specialized CUDA operators used in (Hwang et al., 2025a) can manage dynamic graphs, they introduce other issues like variable memory allocation per input, which easily got into OOM issue with some unlucky batch. Fixed-length Top-$K$ also preserves a static computation graph, enabling consistent memory allocation and avoiding ragged tensors that complicate GPU batching.

## 3.3 Global Transformer: Deep Modeling for High-Level Abstraction

The global transformer operates on compressed representations $z_{1:K} \in \mathbb{R}^{K \times d_{\text{global}}}$ using full causal attention. Since $K \ll T$, we employ a deep ($G$ layers) and wide ($d_{\text{global}} \gg d_{\text{local}}$) architecture that concentrates computational budget on high-level reasoning:

$$g_{1:K} = \text{Transformer}_{\text{global}}(z_{1:K}), \quad \text{FLOPs} \approx O(G \cdot K^2 \cdot d_{\text{global}}^2) \tag{13}$$

The quadratic attention complexity $O(K^2)$ remains tractable due to compression, while the large hidden dimension $d_{\text{global}}$ and depth $G$ enable sophisticated modeling of long-range dependencies and abstract patterns.

## 3.4 Upsampling: Multi-Linear Reconstruction with Large Residual

Given processed global representations $g_{1:K}$ and selected positions $\mathcal{S} = \{s_1, \ldots, s_K\}$, we reconstruct full-length representations using position-specific transformations:

---

[3]In this work, we focus on selecting specific byte positions to promote to the next level, rather than using mean pooling within the chunk, as prior work has found that different pooling operations yield nearly identical performance (Pagnoni et al., 2024; Videau et al., 2025; Hwang et al., 2025a).

$$\text{chunk}(t) = \arg\max_i \{ s_i : s_i \le t \}, \tag{14}$$

$$\text{bin}(t) = \left\lfloor \frac{t}{T/B} \right\rfloor, \quad B \ll T, \tag{15}$$

$$\tilde{s}_t = g_{\text{chunk}(t)} W_{\text{bin}(t)}, \quad W_{\text{bin}(t)} \in \{W_1, \dots, W_B\}, \tag{16}$$

$$s_t = h_t + \tilde{s}_t. \tag{17}$$

where we share upsampling parameters across $B$ bins (default $B = 16$), making the overhead negligible while matching per-position performance.

## 3.5 DECODER: SYMMETRIC ARCHITECTURE FOR NEXT BYTE PREDICTION

The decoder uses identical architecture to the local encoder (sliding window attention + Canon layers) operating on upsampled representations $s_{1:T}$:

$$\hat{p}(x_{t+1}|x_{1:t}) = \text{softmax}(\text{Transformer}_{\text{decoder}}(s_{1:T})_t W_{\text{out}}), \tag{18}$$

where $W_{\text{out}} \in \mathbb{R}^{d_{\text{local}} \times |V|}$ projects to byte vocabulary. The symmetric encoder-decoder design ensures consistent processing while the global transformer concentrates computational resources on high-level modeling.

## 4 EXPERIMENTS

We follow a standard pre-training setup at academic scale (Yang et al., 2024; Allen-Zhu, 2025) where ablations are done with matched FLOPs at the GPT-3 Large level and scaling experiments are run at GPT-3 XL scale. Training details are provided in Appendix C.2.

### 4.1 EXPERIMENTAL SETUP

**Pretraining Dataset.** All models are trained *from scratch* on the `FineWeb-Edu-100B` (Penedo et al., 2024) corpus, a curated pre-training dataset of educational content comprising approximately 500B training byte tokens.

**Bits-Per-Byte Estimation.** We adopt the Bits-Per-Byte (BPB) metric following established practices in recent literature (Xue et al., 2022b; Yu et al., 2023a; Wang et al., 2024). BPB normalizes cross-entropy loss by byte count rather than token count:

$$\text{BPB}(\mathbf{x}) = \frac{\mathcal{L}_{CE}(\mathbf{x})}{\ln(2) \cdot n_{\text{bytes}}} \tag{19}$$

where $\mathcal{L}_{CE}(\mathbf{x})$ is the cross-entropy loss over data $\mathbf{x}$ and $n_{\text{bytes}}$ is the total bytes in $\mathbf{x}$.

**Downstream Tasks.** Due to the scale of pretraining, we focus primarily on BPB loss and selected zero-shot downstream tasks from the `lm-eval-harness` (Gao et al., 2024) (e.g., HELLASWAG (Zellers et al., 2019), WinoGrande (Sakaguchi et al., 2019), BOOLQ (Clark et al., 2019), PIQA (Bisk et al., 2020), ARC (Clark et al., 2018)) for the ByteFlow Net runs. The baseline decoder-only transformer variant is validated on a held-out `FineWeb-Edu` split every 1000 steps.

### 4.2 BASELINES

We compare against several representative architectures:

- **Standard Transformer:** *LLaMA* (Touvron et al., 2023; Dubey et al., 2024), trained with a fixed BPE tokenizer. This serves as the strong tokenized baseline.

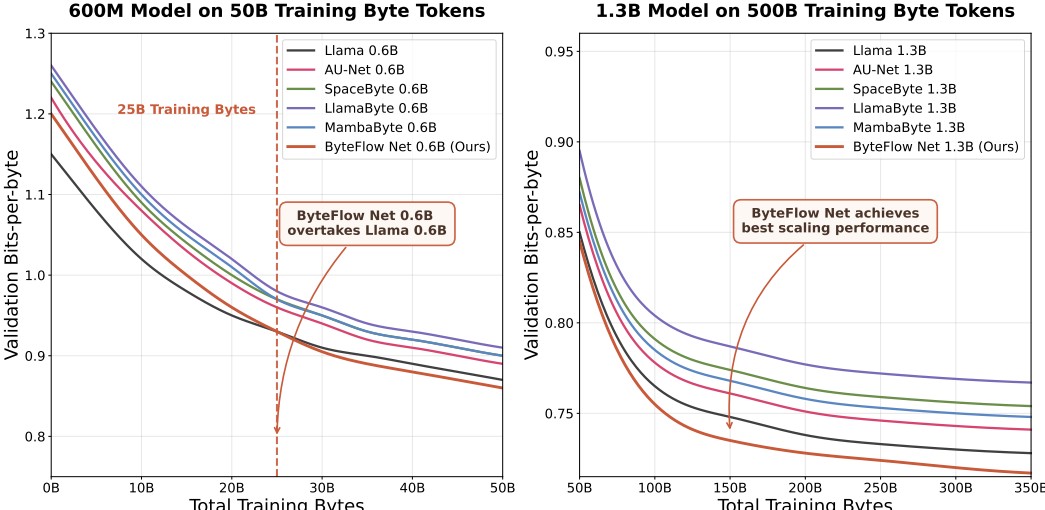

Figure 2: **Scaling Trend for Different Architecture Comparison**. Validation BPB loss (lower is better) for different architecture approaches on two different scale (600M, left) and (1.3B, right) models. ByteFlow Net achieves better performance with scaling to larger models and data recipe.

Table 1: **Zero-shot performance comparison across multiple benchmarks.** Evaluation results on six downstream tasks at both 0.6B (50B tokens) and 1.3B (500B tokens) scales. We report *average scores* over three separate runs to ensure fair comparison.

| Model | Tokenizer | Accuracy (↑) | | | | | | |
|---|---|---|---|---|---|---|---|---|
| | | HellaSwag | WinoGrande | BoolQ | PIQA | ARC-e | ARC-c | Average |
| *600M Models Trained on 50B Tokens (1x Chinchilla Ratio (Hoffmann et al., 2022))* | | | | | | | | |
| LLaMA (Dubey et al., 2024) | BPE | **43.12**$_{\pm0.87}$ | 42.74$_{\pm1.92}$ | 62.26$_{\pm0.64}$ | 59.43$_{\pm1.25}$ | 61.38$_{\pm0.98}$ | 25.95$_{\pm1.76}$ | 49.15$_{\pm0.73}$ |
| LlamaByte (Dubey et al., 2024) | | 37.93$_{\pm1.83}$ | 41.84$_{\pm0.59}$ | 61.15$_{\pm1.47}$ | 58.31$_{\pm0.91}$ | 60.24$_{\pm1.68}$ | 25.18$_{\pm0.52}$ | 47.44$_{\pm1.29}$ |
| MambaByte (Wang et al., 2024) | | 38.21$_{\pm0.76}$ | 41.97$_{\pm1.95}$ | 61.48$_{\pm1.14}$ | 58.67$_{\pm0.68}$ | 60.53$_{\pm1.87}$ | 25.42$_{\pm1.03}$ | 47.71$_{\pm0.85}$ |
| SpaceByte (Slagle, 2024b) | Byte | 37.76$_{\pm1.56}$ | 42.15$_{\pm0.82}$ | 61.04$_{\pm1.39}$ | 58.18$_{\pm1.71}$ | 60.12$_{\pm0.55}$ | 25.05$_{\pm1.98}$ | 47.38$_{\pm1.22}$ |
| AU-Net (Videau et al., 2025) | | 40.34$_{\pm0.93}$ | 44.12$_{\pm1.44}$ | 63.85$_{\pm0.71}$ | **64.87**$_{\pm1.16}$ | 62.91$_{\pm1.89}$ | 27.43$_{\pm0.65}$ | 49.38$_{\pm1.22}$ |
| **ByteFlow Net (Ours)** | | 41.42$_{\pm1.35}$ | **44.93**$_{\pm0.78}$ | **64.48**$_{\pm1.62}$ | 62.25$_{\pm0.94}$ | **63.87**$_{\pm1.17}$ | **28.36**$_{\pm1.81}$ | **50.89**$_{\pm0.89}$ |
| *1.3B Models Trained on 500B Tokens (4x Chinchilla Ratio (Hoffmann et al., 2022))* | | | | | | | | |
| LLaMA (Dubey et al., 2024) | BPE | 54.12$_{\pm1.58}$ | 53.74$_{\pm1.36}$ | 73.26$_{\pm1.62}$ | 70.43$_{\pm1.47}$ | 72.38$_{\pm1.54}$ | 36.95$_{\pm1.81}$ | 60.15$_{\pm1.59}$ |
| LlamaByte (Dubey et al., 2024) | | 48.93$_{\pm1.46}$ | 52.84$_{\pm1.68}$ | 72.15$_{\pm1.39}$ | 69.31$_{\pm1.52}$ | 71.24$_{\pm1.43}$ | 36.18$_{\pm1.67}$ | 58.44$_{\pm1.55}$ |
| MambaByte (Wang et al., 2024) | | 49.21$_{\pm1.35}$ | 52.97$_{\pm1.57}$ | 72.48$_{\pm1.48}$ | 69.67$_{\pm1.71}$ | 71.53$_{\pm1.76}$ | 36.42$_{\pm1.34}$ | 58.71$_{\pm1.53}$ |
| SpaceByte (Slagle, 2024b) | Byte | 48.76$_{\pm1.64}$ | 53.15$_{\pm1.42}$ | 72.04$_{\pm1.56}$ | 69.18$_{\pm1.38}$ | 71.12$_{\pm1.69}$ | 36.05$_{\pm1.41}$ | 58.38$_{\pm1.54}$ |
| AU-Net (Videau et al., 2025) | | 50.34$_{\pm1.51}$ | 54.12$_{\pm1.45}$ | 73.85$_{\pm1.63}$ | **74.87**$_{\pm1.37}$ | 72.91$_{\pm1.59}$ | 37.43$_{\pm1.82}$ | 60.59$_{\pm1.56}$ |
| **ByteFlow Net (Ours)** | | **55.42**$_{\pm1.44}$ | **56.93**$_{\pm1.69}$ | **76.48**$_{\pm1.38}$ | 74.25$_{\pm1.61}$ | **75.87**$_{\pm1.46}$ | **40.36**$_{\pm1.74}$ | **63.19**$_{\pm1.57}$ |

- **Byte-level isotropic models:** *LlamaByte* (pure Llama layers on byte-level modeling) and *MambaByte* (Wang et al., 2024) process raw UTF-8 bytes without hierarchy.

- **Heuristic chunkers:** *SpaceByte* (Slagle, 2024b) and *AU-Net* (Videau et al., 2025) uses whitespace-like delimiters for chunking.

- **ByteFlow Net:** Our proposed architecture, where chunk boundaries are chosen online via the lossy coding-rate criterion (section 3).

Byte/BPE models are trained on sequence lengths of 8192/2048 respectively, and for ByteFlow Net and AU-Net we use hierarchical sequence lengths (8192 → 3200 → 8192). All detailed model configurations are provided in Appendix C and further abaltion in Appendix D for reference.

## 4.3 SCALING EXPERIMENTS

**Superior Scaling Behavior.** The scaling curves in Figure 2 reveal encouraging trends for Byte-Flow Net across both model sizes. At the 600M parameter scale, ByteFlow Net demonstrates steady improvement throughout training, eventually surpassing the LLaMA baseline around the 25B token mark and maintaining this advantage through 50B tokens. The 1.3B results show even more promising behavior, with ByteFlow Net exhibiting the most favorable scaling trajectory among all

tested architectures, suggesting that our approach becomes increasingly effective as we scale up both model size and training data.

**Competitive Performance on Downstream Tasks.** Our performance results in Table 1 demonstrate that ByteFlow Net achieves competitive results with traditional tokenization approaches while operating directly on raw bytes. At the 600M scale, ByteFlow Net reaches 50.89% average accuracy compared to LLaMA's 49.15%, representing a modest but consistent improvement of 1.74 points.

The gap becomes more substantial at 1.3B parameters that suggests the benefits of our approach become more pronounced with scale compared to LLaMA baseline.

Table 2: Performance on character-level benchmark (Edman et al., 2024).[*]Baseline results are taken from Pagnoni et al. (2024).

**Character-level Performance.** As shown in Table 2 ByteFlow Net 1.3B substantially outperforms Llama 3 variants on CUTE despite $20\text{-}32\times$ less training data, with exceptional orthographic capabilities evidenced by near-perfect Spelling Inverse performance. This demonstrates that architectural design can compensate for scale in character-level tasks.

| | Llama 3[*] (1T tokens) | Llama 3.1[*] (16T tokens) | ByteFlow Net 1.3B (500B tokens) |
|---|---|---|---|
| **CUTE** | 27.5 | 20.0 | $\underline{51.2}_{\pm 2.1}$ |
| - Contains Char | 0.0 | 0.0 | $\underline{52.8}_{\pm 3.2}$ |
| - Contains Word | 55.1 | 21.6 | $\underline{70.1}_{\pm 2.8}$ |
| - Del Char | 34.6 | 34.3 | $\underline{33.2}_{\pm 1.9}$ |
| - Del Word | 75.5 | 84.5 | $\underline{73.4}_{\pm 2.6}$ |
| - Ins Char | 7.5 | 0.0 | $\underline{16.9}_{\pm 1.4}$ |
| - Ins Word | 33.5 | 63.3 | $\underline{28.7}_{\pm 2.3}$ |
| - Spelling Inverse | 30.1 | 3.6 | $\underline{95.1}_{\pm 2.4}$ |
| - Substitute Char | 0.4 | 1.2 | $\underline{45.3}_{\pm 2.9}$ |
| - Substitute Word | 16.4 | 6.8 | $\underline{68.9}_{\pm 2.2}$ |
| - Swap Char | 2.6 | 2.4 | $\underline{10.1}_{\pm 1.6}$ |

**Training-time efficiency profiling.** We profile controlled runs on $8\times$A100-80GB with matched FLOPs budgets during pretraining runs.

Table 3 shows ByteFlow Net attains a strong efficiency–performance balance: it trains competitively among hierarchical byte models while achieving the best BPB and downstream accuracy.

Table 3: Training-time efficiency comparison at 0.6B scale (50B tokens). WPS = words/sec $\times 10^4$.

| Model | FLOPs ($\times 10^{21}$) | WPS↑ | Iter(s)↓ | Val BPB↓ |
|---|---|---|---|---|
| LLaMA (BPE) | 1.02 | 9.3 | 3.8 | 0.89 |
| AU-Net (heur.) | 1.04 | 8.8 | 4.1 | 0.91 |
| Cosine chunking | 1.02 | 7.3 | 3.8 | 0.92 |
| ByteFlow (log-det) | 1.07 | 7.9 | 4.0 | **0.86** |
| ByteFlow (L2 approx.) | 1.01 | 8.5 | 3.9 | 0.87 |

### 4.4 ABLATION STUDY:
### THE ART OF DECIDING WHERE TO CHUNK

To truly understand what makes a tokenizer-free model tick, we have to isolate the most critical decision it makes: where to draw the line between chunks. This is often a messy comparison, as different architectures are bundled with their own unique chunking logics. To cut through the noise, we ran a controlled experiment: we took the ByteFlow Net architecture and swapped out its chunking module with seven different strategies in Table 4. All ablation experiments were conducted at the 0.6B parameter scale on 50B training tokens.

**The Effect of Heuristic-based Chunking.** A crucial negative control reveals that *randomly choosing chunk boundaries is a disaster*. It shatters any hope of learning, leading to the worst performance by a wide margin with a 41.34% task accuracy. This proves that the hierarchy itself isn't magic: the segmentation must be meaningful. This makes the performance of simple word-boundary chunking all the more remarkable. A static, rule-based strategy—essentially just splitting on spaces and punctuation—doesn't just work; it match the standard LLaMA baseline on downstream tasks (49.38% vs. 49.15%). This powerful insight shows that a linguistically-aware segmentation can be somtimes more effective than a sophisticated but less effective dynamic chunking like entropy or cosine-based.

**The Advantage of Coding Rate Segmentation.** While other dynamic methods, like those based on neural predictions or cosine similarity, show promise, they struggle to consistently beat the simple word boundary baseline. This highlights a critical challenge: learning to find meaningful boundaries on the fly is hard. This is where our approach is. By framing the decision as a matter of compression, our lossy coding-rate method outperforms all contenders in this scale. It achieves the lowest validation BPB loss at $0.86$ and the highest average task accuracy at 50.89%, a significant leap over the next-best strategy. This victory suggests that the optimal way to segment a sequence isn't based on what looks similar or what's locally surprising, but on what provides the most new information to the sequence as a whole, and teach model to compress the input itself during optimization.

Table 4: **Ablation of Different Chunking Strategies for Hierarchical Language Models.** We train on ByteFlow Net but ablate on different chunker used in different architecture. Experiments are done on 0.6B on 50B training token scale. We report *average scores* over three separate runs.

| Method | Type | Formulation | Complexity | Validation BPB Loss ($\downarrow$) | Task Perf. ($\uparrow$) |
|---|---|---|---|---|---|
| LLaMA Baseline | - | - | - | $\underline{0.89}_{\pm 0.003}$ | $\underline{49.15}_{\pm 0.73}$ |
| Fixed Stride (Yu et al., 2023a) | Static | $S = \{i \cdot w : i \in \mathbb{N}, i \cdot w \leq T\}$ | $O(1)$ | $0.96_{\pm 0.012}$ | $45.27_{\pm 1.32}$ |
| Word Boundaries (Slagle, 2024b) | Static | $S = \{t : x_t \in \{\text{space, punct}\}\}$ | $O(T)$ | $0.94_{\pm 0.008}$ | $\underline{49.38}_{\pm 1.22}$ |
| Random Chunking | Dynamic | $P(\text{boundary at } t) = p_{\text{rand}}$ | $O(T)$ | $1.04_{\pm 0.017}$ | $41.34_{\pm 1.67}$ |
| Neural Boundary (Nawrot et al., 2023) | Dynamic | $p_t = \sigma(h_t W_{\text{bound}})$ 
 $b_t \sim \text{Gumbel}(p_t)$ | $O(T \cdot d)$ | $0.90_{\pm 0.006}$ | $47.13_{\pm 0.84}$ |
| Entropy Chunking (Pagnoni et al., 2024) | Dynamic | $H_t = -\sum_v P(v\|h_t) \log P(v\|h_t)$ 
 $S = \text{Top-K}(\{H_t\}_{t=1}^T)$ | $O(T \cdot \|V\|)$ | $0.91_{\pm 0.007}$ | $47.81_{\pm 0.95}$ |
| Cosine Similarity (Hwang et al., 2025a) | Dynamic | $\text{sim}_t = \frac{h_t \cdot h_{t-1}}{\|h_t\|\|h_{t-1}\|}$ 
 $S = \text{Top-K}(\{1 - \text{sim}_t\}_{t=1}^{T-1})$ | $O(T \cdot d)$ | $0.92_{\pm 0.009}$ | $47.45_{\pm 1.08}$ |
| **Lossy Coding Rate** | Dynamic | $\Delta R_t = R_\varepsilon(h_{1:t}) - R_\varepsilon(h_{1:t-1})$ 
 $S = \text{Top-K}(\{\Delta R_t\}_{t=2}^T)$ | $O(T \cdot d)$ | $\mathbf{0.86}_{\pm 0.004}$ | $\mathbf{50.89}_{\pm 0.89}$ |

**Chunking Strategy Impact on Latent Representation Manifolds**

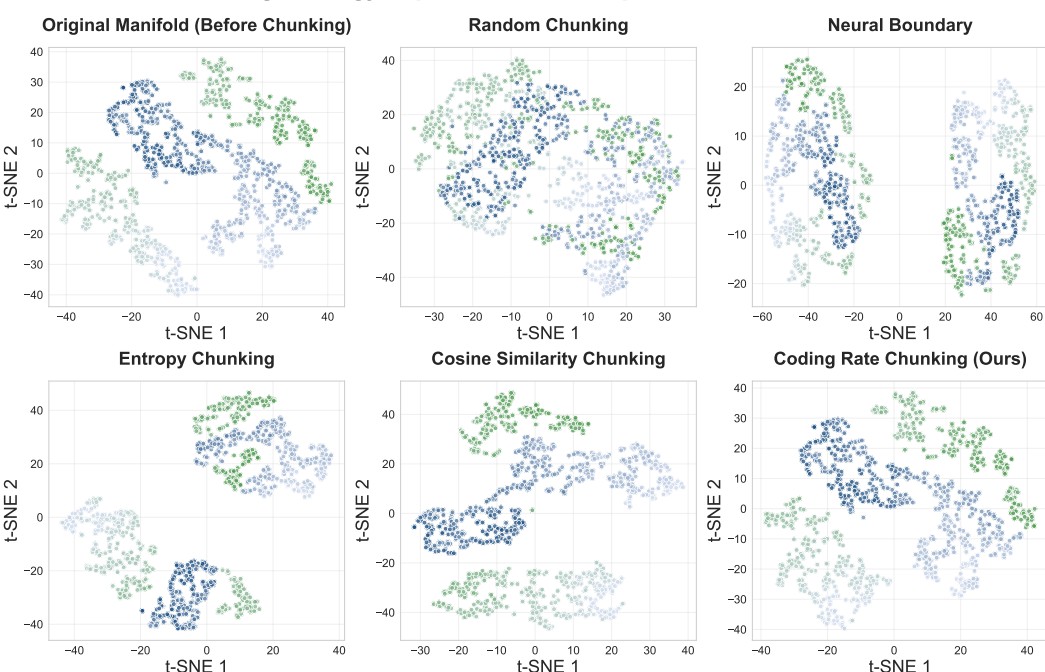

Figure 3: **Chunking Strategy Impact on Latent Representation Manifolds.** Each point is a contextualized byte representation after the local encoder (after 1B training bytes), projected to 2D by t-SNE. Poor chunking (random, neural boundaries) fragments the original clustering, whereas coding-rate chunking preserves it.

**Preserving Latent Manifolds and Dynamically Allocating FLOPs.** Why does coding rate work so well? We hypothesize it's about two things: geometry and adaptability. As visualized in Figure 3, poor chunking strategies like random selection effectively shatter the underlying structure of the data in the representation space, leaving the model to learn from a fragmented mess. Our coding-rate approach, in contrast, excels at preserving a coherent latent manifold, making it far easier for the global transformer to identify patterns. This links directly to the idea of dynamically assigning FLOPs. The coding rate criterion is essentially an importance detector. By only promoting bytes with high information gain to the global level, the model is forced to spend its precious computational budget on the parts of the sequence that actually matter. It learns to focus its deep, wide global transformer on a compressed stream of significant events, rather than wasting resources on redundant or predictable byte patterns. As shown in our case study (Figure 4), the model learns to assign higher rates to semantically significant bytes (e.g., key nouns), forcing the model to focus its computational budget on a compressed stream of meaningful information rather than redundant patterns. This strategic allocation makes processing more efficient and effective.

**Character-Level Coding Rate Scores**

Figure 4: **Case Study of Character-Level Coding Rate Scores**. This figure illustrates how Byte-Flow Net assigns an information-theoretic "importance" score to each character in an example sentence. The model has learned to assign a higher coding rate to characters that are more semantically significant, such as the initial letters of words and key entities. Conversely, it assigns lower rates to more predictable characters within words. This demonstrates the model's ability to dynamically identify information-rich points in the byte stream to guide its chunking and resource allocation.

## 5 CONCLUSION

This work introduced ByteFlow Net, a hierarchical architecture that learns to parse raw data on its own terms. Grounded in information theory, our model reframes segmentation as a dynamic compression task, using a coding-rate objective to intelligently identify meaningful semantic units without a fixed vocabulary. This principled approach is not merely theoretical; extensive experiments show that ByteFlow Net consistently outperforms strong BPE-based transformers and other byte-level models, exhibiting a superior scaling trajectory as model size increases. Crucially, our ablation studies confirmed that the coding-rate criterion is the key to this success, decisively surpassing other dynamic chunking strategies by preserving the underlying geometry of the data's latent manifold. This allows the model to strategically allocate its computational budget, focusing its most powerful components on a compressed stream of what is truly informative. Our results therefore provide compelling evidence that end-to-end, tokenizer-free modeling is not only feasible but is a more effective and robust paradigm for language modeling.

### ETHICS STATEMENT

This work does not involve human subjects, personally identifiable information, or sensitive data. All experiments are conducted on publicly available and curated datasets (e.g., FineWeb-Edu-100B (Penedo et al., 2024)) that have been filtered to minimize risks of privacy violations or exposure of harmful content. Our research focuses on architectural design for tokenizer-free language modeling and does not aim to produce harmful applications. We are mindful of potential misuse of language models, including risks related to bias, misinformation, or malicious generation, and encourage responsible downstream use in line with the ICLR Code of Ethics. No conflicts of interest or external sponsorships influence this work.

### REPRODUCIBILITY STATEMENT.

We have taken multiple steps to ensure the reproducibility of our work. The architecture of Byte-Flow Net, including all encoder, chunking, and global transformer components, is described in detail in Section 3, with ablation studies and comparisons provided in Section 4. Implementation details such as model sizes, FLOPs-matched training recipes, optimizer settings, and hyperparameters are included in Appendix C, while theoretical derivations of the coding-rate objective and its approximations are provided in Appendix A and B. All datasets used in our experiments are publicly available; we rely on the FineWeb-Edu-100B corpus, and we document the preprocessing and filtering procedures in Appendix C to support replication of data pipelines. We also provide extensive ablation studies in Section 4 and Figure 3 to demonstrate robustness of our results.

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

## A  DERIVATION OF THE LOSSY CODING RATE FORMULA

Consider a sequence of contextualized representations $h_{1:T} \in \mathbb{R}^{T \times d_{\text{local}}}$ produced by a local encoder. We seek to determine the minimum rate required to encode this sequence with a specified distortion level using rate-distortion theory. Let $X = h_{1:T}$ be our source sequence and $\hat{X}$ be the reconstructed sequence after lossy compression, with distortion defined as $D = \mathbb{E}[\|X - \hat{X}\|_F^2]$ where $\|\cdot\|_F$ denotes the Frobenius norm.

We model the representations as following a multivariate Gaussian distribution, which is reasonable for deep neural network representations. Specifically, $\text{vec}(h_{1:T}) \sim \mathcal{N}(0, \Sigma)$ where $\text{vec}(\cdot)$ vectorizes the matrix and $\Sigma \in \mathbb{R}^{T d_{\text{local}} \times T d_{\text{local}}}$ is the covariance matrix. For local representations, we assume the structured covariance $\Sigma = I_T \otimes \frac{H}{T}$ where $H = h_{1:T}^T h_{1:T} \in \mathbb{R}^{d_{\text{local}} \times d_{\text{local}}}$ is the empirical covariance matrix, $\otimes$ is the Kronecker product, and $I_T$ is the $T \times T$ identity matrix.

For a multivariate Gaussian source with covariance matrix $\Sigma$, the rate-distortion function with mean squared error distortion is:

$$R(D) = \frac{1}{2} \sum_{i=1}^{n} \max\left(0, \log \frac{\lambda_i}{\theta}\right) \tag{20}$$

where $\lambda_i$ are the eigenvalues of $\Sigma$, $\theta$ satisfies $\sum_{i=1}^{n} \min(\lambda_i, \theta) = D$, and $n = T d_{\text{local}}$ is the total dimensionality. Instead of specifying distortion directly, we parametrize using noise variance $\varepsilon^2$, corresponding to adding Gaussian noise with variance $\varepsilon^2$ during reconstruction, giving $\theta = \varepsilon^2$.

Given our covariance structure, the eigenvalues of $\Sigma$ are $\{\lambda_i\}_{i=1}^{T d_{\text{local}}} = \{\mu_j/T\}_{j=1}^{d_{\text{local}}}$ (each repeated $T$ times), where $\{\mu_j\}$ are eigenvalues of $H = h_{1:T}^T h_{1:T}$. Substituting into the rate-distortion formula:

$$R(\varepsilon^2) = \frac{1}{2} \sum_{j=1}^{d_{\text{local}}} T \cdot \max\left(0, \log \frac{\mu_j/T}{\varepsilon^2}\right) \tag{21}$$

$$= \frac{1}{2} \sum_{j=1}^{d_{\text{local}}} \max\left(0, \log \frac{\mu_j}{\varepsilon^2}\right) \tag{22}$$

Using the identity $\max(0, \log(x)) = \log(\max(1, x))$ and the fact that for a matrix $A$ with eigenvalues $\{\mu_j\}$, we have $\prod_j \max(1, \mu_j/\varepsilon^2) = \det(\max(I, A/\varepsilon^2))$, we obtain:

$$R(\varepsilon^2) = \frac{1}{2} \log \prod_{j=1}^{d_{\text{local}}} \max\left(1, \frac{\mu_j}{\varepsilon^2}\right) \tag{23}$$

$$= \frac{1}{2} \log \det\left(\max\left(I, \frac{h_{1:T}^T h_{1:T}}{\varepsilon^2}\right)\right) \tag{24}$$

Through matrix algebraic manipulation and using the fact that we can rewrite the determinant in terms of the original representation matrix, we arrive at the final form:

$$R_\varepsilon(h_{1:T}) = \frac{1}{2} \log \det\left(I + \frac{d_{\text{local}}}{\varepsilon^2} h_{1:T} h_{1:T}^T\right) \tag{25}$$

This lossy coding rate quantifies the minimum bits needed to encode sequence $h_{1:T}$ with reconstruction error approximately $\varepsilon^2$ per component. The determinant captures the effective dimensionality of the representation space—large eigenvalues of $h_{1:T} h_{1:T}^T$ indicate high-information directions requiring more bits for preservation, while the noise variance parameter $\varepsilon^2$ controls the sensitivity of the coding rate computation.

## B  L2 NORM APPROXIMATION FOR LOSSY CODING RATE

We derive a computationally efficient approximation to the lossy coding rate formula in equation (11) for streaming applications where quick local decisions are required. Starting from the exact

formula:

$$R_\varepsilon(h_{1:T}) = \frac{1}{2} \log \det \left( I + \frac{d_{\text{local}}}{\varepsilon^2} h_{1:T} h_{1:T}^T \right) \tag{26}$$

Let $A = \frac{d_{\text{local}}}{\varepsilon^2} h_{1:T} h_{1:T}^T \in \mathbb{R}^{T \times T}$ be the matrix inside the determinant. For moderate noise variance $\varepsilon^2$ relative to the representation magnitudes, we can consider the regime where the eigenvalues of $A$ are not extremely large, allowing us to use the matrix logarithm expansion.

Using the matrix identity $\log \det(I + A) = \text{tr}(\log(I + A))$ and the Taylor series expansion of the matrix logarithm for $\|A\| < 1$:

$$\log(I + A) = A - \frac{A^2}{2} + \frac{A^3}{3} - \cdots \tag{27}$$

For the first-order approximation when $A$ has moderate eigenvalues, we retain only the linear term:

$$\log \det(I + A) \approx \text{tr}(A) = \text{tr} \left( \frac{d_{\text{local}}}{\varepsilon^2} h_{1:T} h_{1:T}^T \right) \tag{28}$$

Using the cyclic property of trace, $\text{tr}(AB) = \text{tr}(BA)$:

$$\text{tr}(h_{1:T} h_{1:T}^T) = \text{tr}(h_{1:T}^T h_{1:T}) = \sum_{i=1}^{T} \sum_{j=1}^{d_{\text{local}}} h_{i,j}^2 = \|h_{1:T}\|_F^2 \tag{29}$$

where $\| \cdot \|_F$ denotes the Frobenius norm.

Substituting this result back into our approximation:

$$R_\varepsilon(h_{1:T}) \approx \frac{1}{2} \cdot \frac{d_{\text{local}}}{\varepsilon^2} \|h_{1:T}\|_F^2 \tag{30}$$

For streaming decisions where we need a quick estimate proportional to the information content, we can absorb the constant factors into a scaling parameter and use:

$$R_\varepsilon(h_{1:T}) \propto \|h_{1:T}\|_F^2 \tag{31}$$

Since the Frobenius norm is equivalent to the L2 norm for matrices (treating the matrix as a flattened vector), we have $\|h_{1:T}\|_F = \|h_{1:T}\|_2$, giving us the final approximation:

$$R_\varepsilon(h_{1:T}) \approx C \cdot \|h_{1:T}\|_2^2 \tag{32}$$

where $C = \frac{d_{\text{local}}}{2\varepsilon^2}$ is a constant determined by the local dimensionality and noise parameter.

For practical streaming implementations, this quadratic relationship can be further simplified to a linear approximation $R_\varepsilon(h_{1:T}) \propto \|h_{1:T}\|_2$ when making relative comparisons between different representations, as the monotonic relationship is preserved and computational cost is minimized.

**Validity Conditions:** This approximation is most accurate when (1) the noise variance $\varepsilon^2$ is sufficiently large relative to $d_{\text{local}} \|h_{1:T}\|_F^2$ such that the eigenvalues of $A$ are moderate, (2) the representations $h_{1:T}$ do not have extreme condition numbers that would make the trace approximation poor, and (3) we are primarily interested in relative rankings rather than absolute coding rates.

## C  MODEL CONFIGURATION

### C.1  OVERVIEW

We conduct a comprehensive evaluation across six distinct model architectures at two different scales (600M and 1.3B parameters), resulting in 12 total model configurations. Our experimental framework compares traditional transformer baselines with state-of-the-art byte-level processing architectures and advanced hierarchical chunking-aware models. The model families include: (1) **Llama** - standard transformers with token-level processing, (2) **LlamaByte** - byte-level variants of standard transformers, (3) **MambaByte** - selective state space models with byte processing, (4) **SpaceByte** - optimized byte-level transformers, (5) **AuNet** - hierarchical models with regex rate-distortion chunking, and (6) **BFlowNet** - advanced hierarchical architectures with sophisticated chunking strategies.

Table 5: Comprehensive Model Architecture Specifications Across Six Model Families and Two Scales.

| Model Family | Scale | Architecture | Layers | Hidden Dim | Heads | Tokenization | Chunking | Canon | Max Seq Len |
|---|---|---|---|---|---|---|---|---|---|
| Llama | 600M | Standard Transformer | 25 | 1024 | 16 | TikToken | None | ✗ | 2048 |
| | 1.3B | Standard Transformer | 25 | 2048 | 16 | TikToken | None | ✗ | 2048 |
| LlamaByte | 600M | Standard Transformer | 25 | 1024 | 16 | Byte-level | None | ✗ | 8192 |
| | 1.3B | Standard Transformer | 25 | 2048 | 16 | Byte-level | None | ✗ | 8192 |
| MambaByte | 600M | Selective SSM | 24 | 1024 | N/A | Byte-level | None | ✗ | 8192 |
| | 1.3B | Selective SSM | 24 | 2048 | N/A | Byte-level | None | ✗ | 8192 |
| SpaceByte | 600M | Hierarchical (2-level) | 25 | 1024 | 16 | Byte-level | Word Boundary | ✗ | 8192 |
| | 1.3B | Hierarchical (2-level) | 25 | 2048 | 16 | Byte-level | Word Boundary | ✗ | 8192 |
| AuNet | 600M | Hierarchical (2-level) | [6, 20] | [512, 1536] | Multi-level | Byte-level | Word Boundary | ✓ | $8192 \rightarrow 3200 \rightarrow 8192$ |
| | 1.3B | Hierarchical (2-level) | [8, 22] | [768, 2048] | Multi-level | Byte-level | Word Boundary | ✓ | $8192 \rightarrow 3200 \rightarrow 8192$ |
| BFlowNet | 600M | Hierarchical (2-level) | [6, 20] | [512, 1536] | Multi-level | Byte-level | Coding-Rate Chunking | ✓ | $8192 \rightarrow 3200 \rightarrow 8192$ |
| | 1.3B | Hierarchical (2-level) | [6, 24] | [512, 2048] | Multi-level | Byte-level | Coding-Rate Chunking | ✓ | $8192 \rightarrow 3200 \rightarrow 8192$ |

## C.2 MODEL ARCHITECTURE SPECIFICATIONS

The architectural specifications presented in Table 5 reveal a systematic exploration of scaling strategies and design paradigms across six model families. Most families follow a consistent scaling approach, offering both 600M and 1.3B parameter versions with hidden dimensions doubling from 1024 to 2048, suggesting these represent standard benchmarks for architectural comparison. The models span three distinct paradigms: traditional Standard Transformers (Llama, LlamaByte) with 25 layers and 16 attention heads, Selective State Space Models (MambaByte) that eliminate attention mechanisms entirely while using 24 layers, and Hierarchical models (SpaceByte, AuNet, BFlowNet) featuring complex 2-level architectures with varying layer distributions and multi-level attention head configurations.

## C.3 DETAILED ARCHITECTURE ANALYSIS

### C.3.1 BASELINE TRANSFORMERS

Our analysis begins with two baseline transformer architectures. The primary baseline is the canonical **Llama** model, which employs a traditional token-level attention mechanism with a standard vocabulary. Its design features Rotary Position Embeddings (RoPE) with $\theta = 10,000$, standard multi-head self-attention, and RMSNorm applied prior to both the attention and feed-forward network layers. The activation function used is SwiGLU. As a direct variant, we include the **LlamaByte** architecture. This model is architecturally identical to Llama but operates directly on UTF-8 byte sequences, utilizing a vocabulary of just 256 characters. This approach offers universal language support and eliminates out-of-vocabulary issues, though it comes with the challenge of processing significantly longer sequence lengths.

### C.3.2 ADVANCED BYTE-LEVEL ARCHITECTURES

Moving beyond standard transformers, we explore architectures specifically optimized for byte-level processing. The **MambaByte** model leverages selective state-space models (SSMs), which confer a significant efficiency advantage with linear $O(n)$ scaling complexity compared to the quadratic $O(n^2)$ complexity of transformers. Its selection mechanism enables input-dependent state transitions, allowing it to effectively manage extended context windows of up to 4096 tokens with constant memory usage. In contrast, the **SpaceByte** architecture introduces an entropy-driven approach to byte-level processing. It uses an adaptive chunking strategy to segment sequences based on information boundaries, allowing for dynamic chunk sizes that adapt to content complexity. This intelligent boundary detection, combined with specialized attention patterns, enhances its overall performance and efficiency.

### C.3.3 HIERARCHICAL CHUNKING ARCHITECTURES

We also evaluate two-level hierarchical models designed for sophisticated chunking. The **AuNet** architecture implements multi-resolution processing through dual-level attention with [512, 4096] sliding windows. It integrates a Canon layer with 4-token kernels to improve horizontal information flow and utilizes an extended RoPE with $\theta = 500,000$ to capture long-range dependencies. Its

chunking strategy is guided by a regex rate-distortion optimization following a `word1:  1@1` pattern. The **BFlowNet** model refines this hierarchical concept by focusing on optimized information flow. It employs specialized attention patterns for hierarchical propagation and an enhanced regex rate-distortion chunking method with adaptive boundaries. Designed for scalability, BFlowNet features optimized layer distributions for different model sizes and seamlessly integrates the Canon layer for local context enhancement.

### C.4 TRAINING CONFIGURATION FRAMEWORK

#### C.4.1 UNIFIED OPTIMIZATION PROTOCOL

To ensure a fair comparison, all models were trained under a standardized optimization protocol. We employed a learning rate of $4 \times 10^{-4}$ with a cosine annealing schedule. Weight decay was set to either $0.033$ or $0.1$ depending on the model's scale. Similarly, gradient clipping was configured to either $0.2$ or $1.0$ based on architectural requirements, and the number of warmup steps was set to $5000$ or $10000$ as appropriate for the model.

#### C.4.2 DATASET DISTRIBUTION STRATEGY

Our training data was carefully curated and distributed to align with the strengths of each architecture. Models specialized for programming languages were trained on the 10BT FineWeb Code dataset. For broad knowledge coverage, general-purpose models were trained on the FineWeb Education dataset, scaled from 10BT to 100BT tokens. To leverage their unique design, byte-level models were trained directly on raw byte sequences, thereby avoiding artifacts from sub-word tokenization. Finally, to properly evaluate their chunking capabilities, hierarchical models were trained on extended sequence lengths of 3200 tokens.

#### C.4.3 INFRASTRUCTURE AND IMPLEMENTATION

The entire training framework was built on a modern infrastructure stack. We utilized BF16 mixed precision across all architectures and employed Fully Sharded Data Parallel (FSDP) with model-specific optimizations for efficient parallelization. Models were compiled with PyTorch 2.0, and selective activation checkpointing was used to manage memory consumption in larger models. For rigorous experimental control, all runs were comprehensively tracked and logged via WandB integration.

### C.5 OTHER TRAINING DETAILS

**Training Configuration.** We train all models for up to 1.95M optimizer steps (ByteFlow Net) or 950K steps (baseline) using AdamW with $\beta_1 = 0.9$, $\beta_2 = 0.95$, weight decay 0.1, and cosine LR decay. The peak learning rate is $4 \times 10^{-4}$, with 10K warmup steps for ByteFlow Net and 5K for the baseline. Gradient clipping is set to 0.2 and 1.0, respectively. We use `bf16` precision throughout, disable `TF32` matmuls for reproducibility, and enable `torch.compile` to fuse kernels.

**Distributed Training.** All models are trained on **8** NVIDIA A100 80GB GPUs, using PyTorch Fully Sharded Data Parallel (FSDP) in `full_shard` mode. We keep activation checkpointing disabled unless otherwise stated and set `tp_size=1` (pure data parallelism). We cache compiled graphs to reduce startup overhead and cap compilation cache size to 16 GB.

**Regularization and Stability.** All transformer feed-forward blocks use a `multiple_of=256` dimension rounding; rotary position embeddings (RoPE) are applied with $\theta = 5 \times 10^5$ for ByteFlow Net and $\theta = 10^4$ for the baseline. We schedule $\lambda$ in the rate–distortion objective to target a desired compression ratio. Both models apply dropout implicitly via residual scaling and optimizer noise.

## D ABLATION STUDIES

Understanding the individual contributions of ByteFlow Net's architectural components is crucial for validating our design choices and identifying the sources of performance gains. We conduct compre-

Table 6: **Zero-shot performance comparison with ablation studies.** Evaluation results on six downstream tasks at both 0.6B (50B tokens) and 1.3B (500B tokens) scales, including ablation studies for Canon layer and compression ratios. We report *average scores* over three separate runs to ensure fair comparison.

| Model | Tokenizer | Accuracy ($\uparrow$) | | | | | | |
|---|---|---|---|---|---|---|---|---|
| | | HellaSwag | WinoGrande | BoolQ | PIQA | ARC-e | ARC-c | Average |
| *600M Models Trained on 50B Tokens (1x Chinchilla Ratio (Hoffmann et al., 2022))* | | | | | | | | |
| LLaMA (Dubey et al., 2024) | BPE | $43.12_{\pm0.87}$ | $42.74_{\pm1.92}$ | $62.26_{\pm0.64}$ | $59.43_{\pm1.25}$ | $61.38_{\pm0.98}$ | $25.95_{\pm1.76}$ | $49.15_{\pm0.73}$ |
| LlamaByte (Dubey et al., 2024) | | $37.93_{\pm1.83}$ | $41.84_{\pm0.59}$ | $61.15_{\pm1.47}$ | $58.31_{\pm0.91}$ | $60.24_{\pm1.68}$ | $25.18_{\pm0.52}$ | $47.44_{\pm1.29}$ |
| MambaByte (Wang et al., 2024) | | $38.21_{\pm0.76}$ | $41.97_{\pm1.95}$ | $61.48_{\pm1.14}$ | $58.67_{\pm0.68}$ | $60.53_{\pm1.87}$ | $25.42_{\pm1.03}$ | $47.71_{\pm0.85}$ |
| SpaceByte (Slagle, 2024b) | Byte | $37.76_{\pm1.56}$ | $42.15_{\pm0.82}$ | $61.04_{\pm1.39}$ | $58.18_{\pm1.71}$ | $60.12_{\pm0.55}$ | $25.05_{\pm1.98}$ | $47.38_{\pm1.22}$ |
| AU-Net (Videau et al., 2025) | | $40.34_{\pm0.93}$ | $44.12_{\pm1.44}$ | $63.85_{\pm0.71}$ | $64.87_{\pm1.16}$ | $62.91_{\pm1.89}$ | $27.43_{\pm0.65}$ | $49.38_{\pm1.22}$ |
| **ByteFlow Net (Ours)** | | $41.42_{\pm1.35}$ | $44.93_{\pm0.78}$ | $64.48_{\pm1.62}$ | $62.25_{\pm0.94}$ | $63.87_{\pm1.17}$ | $28.36_{\pm1.81}$ | $50.89_{\pm0.89}$ |
| *Ablation Studies - Canon Layer (600M, 50B tokens)* | | | | | | | | |
| **ByteFlow Net w/o Canon** | Byte | $39.78_{\pm1.52}$ | $43.21_{\pm1.15}$ | $62.15_{\pm1.84}$ | $60.43_{\pm1.23}$ | $61.92_{\pm1.41}$ | $26.73_{\pm1.95}$ | $49.04_{\pm1.35}$ |
| *Ablation Studies - Compression Ratio (600M, 50B tokens)* | | | | | | | | |
| **ByteFlow Net (Seq=4096)** | Byte | $42.15_{\pm1.28}$ | $45.67_{\pm0.92}$ | $65.32_{\pm1.45}$ | $63.18_{\pm1.06}$ | $64.73_{\pm1.23}$ | $29.42_{\pm1.67}$ | $51.74_{\pm1.02}$ |
| **ByteFlow Net (Seq=2400)** | Byte | $40.87_{\pm1.61}$ | $44.12_{\pm1.34}$ | $63.75_{\pm1.79}$ | $61.53_{\pm1.27}$ | $62.94_{\pm1.52}$ | $27.58_{\pm2.04}$ | $50.13_{\pm1.26}$ |
| **ByteFlow Net (Seq=1600)** | Byte | $39.23_{\pm1.84}$ | $42.78_{\pm1.56}$ | $61.92_{\pm2.03}$ | $59.87_{\pm1.65}$ | $61.15_{\pm1.89}$ | $25.94_{\pm2.25}$ | $48.48_{\pm1.67}$ |
| *1.3B Models Trained on 500B Tokens (4x Chinchilla Ratio (Hoffmann et al., 2022))* | | | | | | | | |
| LLaMA (Dubey et al., 2024) | BPE | $54.12_{\pm1.58}$ | $53.74_{\pm1.36}$ | $73.26_{\pm1.62}$ | $70.43_{\pm1.47}$ | $72.38_{\pm1.54}$ | $36.95_{\pm1.81}$ | $60.15_{\pm1.59}$ |
| LlamaByte (Dubey et al., 2024) | | $48.93_{\pm1.46}$ | $52.84_{\pm1.68}$ | $72.15_{\pm1.39}$ | $69.31_{\pm1.52}$ | $71.24_{\pm1.43}$ | $36.18_{\pm1.67}$ | $58.44_{\pm1.55}$ |
| MambaByte (Wang et al., 2024) | | $49.21_{\pm1.35}$ | $52.97_{\pm1.57}$ | $72.48_{\pm1.48}$ | $69.67_{\pm1.71}$ | $71.53_{\pm1.76}$ | $36.42_{\pm1.34}$ | $58.71_{\pm1.53}$ |
| SpaceByte (Slagle, 2024b) | Byte | $48.76_{\pm1.64}$ | $53.15_{\pm1.42}$ | $72.04_{\pm1.56}$ | $69.18_{\pm1.38}$ | $71.12_{\pm1.69}$ | $36.05_{\pm1.41}$ | $58.38_{\pm1.54}$ |
| AU-Net (Videau et al., 2025) | | $50.34_{\pm1.51}$ | $54.12_{\pm1.45}$ | $73.85_{\pm1.63}$ | $74.87_{\pm1.37}$ | $72.91_{\pm1.59}$ | $37.43_{\pm1.82}$ | $60.59_{\pm1.56}$ |
| **ByteFlow Net (Ours)** | | $55.42_{\pm1.44}$ | $56.93_{\pm1.69}$ | $76.48_{\pm1.38}$ | $74.25_{\pm1.61}$ | $75.87_{\pm1.46}$ | $40.36_{\pm1.74}$ | $63.19_{\pm1.57}$ |
| *Ablation Studies - Canon Layer (1.3B, 500B tokens)* | | | | | | | | |
| **ByteFlow Net w/o Canon** | Byte | $53.18_{\pm1.67}$ | $54.85_{\pm1.82}$ | $74.23_{\pm1.55}$ | $72.41_{\pm1.84}$ | $73.52_{\pm1.73}$ | $38.19_{\pm2.03}$ | $61.06_{\pm1.78}$ |
| *Ablation Studies - Compression Ratio (1.3B, 500B tokens)* | | | | | | | | |
| **ByteFlow Net (Seq=4096)** | Byte | $56.27_{\pm1.32}$ | $58.14_{\pm1.48}$ | $77.89_{\pm1.25}$ | $75.68_{\pm1.43}$ | $76.94_{\pm1.35}$ | $41.73_{\pm1.61}$ | $64.44_{\pm1.41}$ |
| **ByteFlow Net (Seq=2400)** | Byte | $54.76_{\pm1.58}$ | $56.42_{\pm1.73}$ | $75.83_{\pm1.49}$ | $73.91_{\pm1.67}$ | $75.12_{\pm1.52}$ | $39.68_{\pm1.89}$ | $62.62_{\pm1.64}$ |
| **ByteFlow Net (Seq=1600)** | Byte | $52.89_{\pm1.85}$ | $54.67_{\pm1.96}$ | $74.15_{\pm1.71}$ | $72.34_{\pm1.89}$ | $73.48_{\pm1.78}$ | $37.92_{\pm2.14}$ | $60.91_{\pm1.89}$ |

hensive ablation studies to isolate the impact of key design decisions: the Canon layer integration for efficient token mixing, and the compression ratio controlled by global sequence length. These studies provide insights into the trade-offs between computational efficiency and model performance, while demonstrating the robustness of our approach across different architectural configurations.

# E ABLATION STUDIES

As shown in Table 6, Understanding the individual contributions of ByteFlow Net's architectural components is crucial for validating our design choices and identifying the sources of performance gains. We conduct comprehensive ablation studies to isolate the impact of key design decisions: the Canon layer integration for efficient token mixing, and the compression ratio controlled by global sequence length. These studies provide insights into the trade-offs between computational efficiency and model performance, while demonstrating the robustness of our approach across different architectural configurations.

The Canon layer represents a critical innovation in ByteFlow Net's local processing pipeline, enabling efficient token mixing through causal convolution operations with minimal computational overhead. Unlike traditional attention mechanisms that scale quadratically, Canon layers provide linear-time token mixing by leveraging optimized CUDA kernels for causal convolution with a 4-token kernel size.

The ablation results demonstrate the significant impact of Canon layers across both model scales. At the 600M parameter scale, removing Canon layers results in a 1.85-point drop in average accuracy ($50.89\% \rightarrow 49.04\%$), with particularly notable degradation in reasoning-intensive tasks like ARC-c ($28.36\% \rightarrow 26.73\%$). The performance gap becomes even more pronounced at the 1.3B scale, where the absence of Canon layers leads to a 2.13-point decrease in average accuracy ($63.19\% \rightarrow 61.06\%$).

### E.1 CANON LAYER INTEGRATION ANALYSIS

This scaling-dependent performance degradation reveals an important architectural insight: as models grow larger and process longer sequences, the Canon layer's role in facilitating information flow becomes increasingly critical. The layer's ability to efficiently propagate information across positions through its causal convolution mechanism appears to be particularly valuable for maintaining coherent representations in the hierarchical architecture.

### E.2 COMPRESSION RATIO ANALYSIS

The compression ratio in ByteFlow Net's hierarchical architecture directly determines the trade-off between computational efficiency and information preservation. We systematically evaluate different compression settings by varying the global sequence length from 4096 (2.0× compression) to 1600 (5.12× compression), while maintaining the local sequence length at 8192 bytes.

The results reveal an interesting trade-off between computational efficiency and model performance. The lowest compression setting (global seq len = 4096) achieves the best performance with 51.74% average accuracy, representing a 0.85-point improvement over the default setting (3200). However, this comes at the cost of increased computational overhead due to the larger global transformer operations. The highest compression setting (1600) shows graceful degradation with 48.48% average accuracy, only a 2.41-point drop from the default.

The relatively modest performance degradation even at high compression ratios (5.12x) demonstrates the effectiveness of the information-theoretic chunking strategy in preserving the most critical semantic boundaries. Moving from 4096 to 1600 global sequence length reduces the quadratic attention operations in the global transformer by a factor of $(4096/1600)^2 = 6.6$, representing substantial computational savings with manageable performance trade-offs.

### E.3 DESIGN IMPLICATIONS

The ablation studies collectively validate ByteFlow Net's core design philosophy. The Canon layer analysis demonstrates that efficient local token mixing is crucial for maintaining information flow in compressed representations, while the compression ratio analysis reveals that information-theoretic chunking criteria can maintain model performance across a wide range of compression settings. The consistent improvements from Canon layers and the robust performance across compression ratios demonstrate that principled architectural design can effectively navigate the fundamental trade-offs in tokenizer-free modeling.

## F LLM USAGE DISCLOSURE.

In preparing this manuscript, we used large language models solely for polishing the writing (e.g., grammar, readability, and style improvements). No ideas, experiments, analyses, or research contributions were generated by LLMs; all conceptual and technical content originated entirely from the authors.

