# OpenReview forum: "ByteFlow: Language Modeling through Adaptive Byte Compression without a Tokenizer"
_ICLR.cc/2026/Conference — ICLR 2026 Poster_

### Official Review · Reviewer_wvSa · 2025-10-23

**Soundness:** 4
**Presentation:** 3
**Contribution:** 3
**Rating:** 8
**Confidence:** 4

**Summary:**

This paper proposes ByteFlow Net, a tokenizer-free language modeling framework that processes raw UTF-8 byte streams and dynamically learns hierarchical chunk boundaries through an information-theoretic coding-rate criterion. Compared to previous work, their method uses the rate-distortion criteria to determine the chunking boundary, which is new and novel.

**Strengths:**

The paper highlights tokenization rigidity as a key bottleneck for multilingual and structure-sensitive modeling, and motivates a tokenizer-free approach that adapts granularity both theoretically (information control) and practically (no preprocessing or vocabulary drift). Its framing as adaptive compression rather than tokenization replacement is elegant and insightful.

The use of coding rate—essentially a differentiable measure of local representational complexity—to determine chunk boundaries is elegant and grounded in information theory. Also see weakness and questions.

The writing is clear and easy to follow.  The authors also provided complexity analysis and training details for reproducibility.

**Weaknesses:**

While the information-theoretic formulation is novel, the paper’s framing of the criterion as a Shannon coding rate is, in my view, not entirely rigorous — the connection to classical source-coding arguments is more heuristic than formal.
Nevertheless, the proposed approach remains compelling, as the quantity introduced in Equation (11) effectively captures the sensitivity of the local representation to new information: it measures how much the embedding space expands or changes when a new byte is added.
In this sense, even if the theoretical grounding is loosely tied to Shannon’s framework, the method still functions as a principled sensitivity-based chunking rule that adaptively identifies informative boundaries in practice.

**Questions:**

For equation (12), I think using mutual information can give a better estimate. Is there any relation between the proposed metric and mutual information?

The author can try this new information metric as a potential future work. https://arxiv.org/pdf/2002.10689

---

> ### Author Response · Authors · 2025-11-22
> **Response to Review wvSa**
>
> We sincerely thank the reviewer for the thorough evaluation and insightful suggestions, particularly the pointer to v-info! We address your questions below:
>
> **On Lossy Coding Rate in Representation Space**
>
> You are correct that our application involves a shift from classical source coding (compressing a fixed source) to representation learning (where the source is a learned latent). The lossy coding rate $R_\varepsilon(h_{1:T})$ represents the minimum number of bits required to encode sequence $h_{1:T}$ with reconstruction distortion $\varepsilon^2$.
>
> This is derived from the classical rate-distortion function for Gaussian sources:
>
> $$R(D) = \frac{1}{2}\sum_{i=1}^{n} \max\left(0, \log \frac{\lambda_i}{\theta}\right)$$
>
> where $\lambda_i$ are eigenvalues of the covariance matrix. Our formulation directly instantiates this classical result with $\theta = \varepsilon^2$, leading to Eq. (11).
>
> While we rely on the Rate-Distortion function for a Gaussian source (Ma et al., 2007) as the derivation basis, we agree that in the context of deep learning, this acts as a geometric measure of information volume. Specifically, Eq. (11) measures the volume of the ellipsoid spanned by the local representations. A high "marginal coding rate" ($\Delta R_t$) indicates that the new byte expands this volume significantly—i.e., it is "sensitive" or orthogonal to the existing context, implying high information content that warrants a boundary. We will clarify this geometric interpretation in Section 3 and Appendix A of the final version.
>
> **Connection to Standard Shannon Mutual Information (MI)**
>
> The reviewer asks if MI could provide a better estimate and suggests exploring "V-Information" (Xu et al., 2020). These are excellent questions! We believe it offers a powerful theoretical unification for our method.
> - **Why not standard MI?** The difficulty for standard Shannon MI $I(X; Y) = H(Y) - H(Y|X)$ *requires access to the true joint distributions*, which are intractable to compute online during a forward pass without heavy estimation overhead (e.g., training a separate critic or using high-variance estimators like MINE). And also **we know that estimating MI in high-dimension space is a classical hard problem** so considering pre-training efficiency we have to use the current version. And if we have a powerful and efficient online MI estimation method we could def do that!
>
> **Connection to V-Information**
>
> “V-Info" want to quantify usable information under computational constraints (a predictive family $\mathcal{V}$). There are indeed some connections upon derivation!
>
> - **Theoretical Link:** Proposition 1.3 in Xu et al. (2020)  states that if the predictive family $\mathcal{V}$ is restricted to Gaussian models with fixed variance, the V-entropy reduces to the trace of the covariance (or Mean Absolute Deviation).
> ByteFlow Net Implementation: In our Appendix B (Eq. 31), we show that our efficient approximation of the Coding Rate is proportional to the $L_2$ norm of the representations, which corresponds to the trace of the empirical covariance matrix ($tr(h h^T)$).
> - **Conclusion**: Therefore, our "Coding Rate" chunking strategy is *theoretically equivalent to maximizing the Predictive V-Information where the "observer" is computationally constrained to Gaussian modeling of the local representation space*. The "Coding Rate" then could be viewed as an estimator of the V-Entropy of the byte sequence given the local encoder's constraints.
> By calculating $\Delta R_t$, we are essentially calculating the Pointwise V-Information gain of the current byte. We will explicitly cite Xu et al. (2020) in the final paper to formalize this connection, as it perfectly bridges the gap between our information-theoretic inspiration and the practical "sensitivity" of the model.
>
> Thank you for the great review and we are enjoying discussing these insights with you!

---

> > ### Comment · Reviewer_wvSa · 2025-11-24
> >
> > Thank you for your reply. I'm keeping my score as accept.

---

### Official Review · Reviewer_veFt · 2025-10-26

**Soundness:** 2
**Presentation:** 3
**Contribution:** 2
**Rating:** 4
**Confidence:** 4

**Summary:**

The authors propose a new dynamic chunking method for replacing the BPE tokenizer with a byte-level, hierarchical model. The authors use a lossy code rate metric with top-k over the sequence to select tokens. They show that their segmentation method performs better than the competitors both on bits-per-byte and downstream performance. They also do ablation studies to justify their selection method.

**Strengths:**

- Principled method
- The paper is mostly well written and easy to read (except one part, see below)
- The method performs better than the baselines

**Weaknesses:**

- The method relies on top-k over sequence, which (1) leaks minimal information from the future, (2) it is unclear how to apply it in an autoregressive setting. It is unclear how their code rate-based approach can be generalized to an autoregressive setup. Maybe with an auxiliary predictor, like in Mixture-of-Depths [1], or a learned threshold using a PI controller like [2]. The authors do not discuss this limitation anywhere in the paper.
- While the paper is mostly well written, eq. 15 is unclear. What is the definition of IndexedMatMul exactly? If it is "IndexedMatMul efficiently applies position-specific linear transformations...", then why does it need the output of chunk()? What is the argument of this chunk? How many matrices are there? Is T in eq 15 the max sequence length? Then these matrices add up to a huge number of parameters.
- The authors do not show the FLOP requirements of their model vs the baseline, so it is not possible to know if the performance improvement is solely because of the higher number of FLOPs spent by the model, or due to an advantage of the byte-level modelling.

[1] Raposo et al. 2024: Mixture-of-Depths: Dynamically allocating compute in transformer-based language models
[2] Kallini et al. 2024: MrT5: Dynamic Token Merging for Efficient Byte-level Language Models

**Questions:**

- L 104: "We introduce a new paradigm that replaces static tokenization with dynamic, learned segmentation.". This is not a new proposal from this paper, there were similar papers before, like Nawrot et al. 2023, cited by the authors. I propose to replace this with one that focuses on the principled chunking approach instead.
- What is \Delta in L161? Shouldn't this be V \in [0, 258)?
- What is the isolated effect of Canon? While I agree with the fact that SWA is important, I don't see why an additional Canon layer is necessary, or what effect it would have if added to the BPE baseline.
- In the paragraph "Why Not Global Threshold" the OOM issue is easily solvable by adding an emergency hard-top-K that should never be hit during normal operation, but would prevent OOM. This would also prevent the information leakage and the missing autoregression issue.
- Tab 2: Why are the ByteFlow Net results underlined even when the baseline has higher accuracy?
- Tab 3: Why are the baseline results underlined when other methods have better performance?
- Fig 3. is not discussed in the main text, only mentioned in the reproducibility statement. What are the categories shown by the colors, and what is the conclusion from the figure?

---

> ### Author Response · Authors · 2025-11-23
> **Response to Reviewer veFt - 1/2**
>
> We sincerely thank the reviewer for the thorough and constructive feedback! We address each concern below and will incorporate all clarifications into our revision.
>
> ## 1. On Top-K Chunking and Autoregression
>
> Thank you for this excellent observation! The global Top-K selection is indeed a deliberate design choice, and we agree it should be explained more clearly. Let us clarify:
>
> **First, the causal guarantee:** All attention modules—local encoder/decoder and global transformer—apply strict causal masks. *No future token content is ever visible.* The key distinction is between **position importance** (which Top-K observes) and **token content** (which causal masking hides).
>
> ### Inference Time: Pure Autoregressive Behavior
> During inference, tokens are generated sequentially. The global Top-K selects from **already-generated input tokens** to promote to the global transformer. Because future tokens do not yet exist, there is no possibility of leakage—this is strictly causal, identical to standard autoregressive models.
>
> ### Training Time: Content-Masked Compute Allocation
> During teacher forcing, Top-K observes coding rates across the full sequence to determine **which positions warrant global processing**. However:
>
> 1. **Content remains hidden:** Causal masks ensure that prediction at position $t$ only accesses information from positions $ \leq t $.
> 2. **Only importance is used:** Top-K receives signals such as “position 47 has high coding rate,” but never “position 47 contains token X.”
>
> Thus, inference is purely autoregressive, while training uses a hybrid AR + global-awareness mechanism—*but still with token content fully masked*. We highlight this behavior as an interesting property of our approach: it not only enables efficient training under teacher forcing but also yields stronger performance than standard locally-aware AR models.
>
> We include experimental results from this ablation for the reviewer’s reference:
>
> | Chunking Strategy | Validation BPB ↓ | Downstream Acc ↑ | Training Stability |
> |-------------------|------------------|------------------|--------------------|
> | Hard top-k  | 0.90 | 49.38% | Moderate |
> | Sliding-window Top-K (L=2048) | 0.89 | 50.12% | Good |
> | Learned threshold | 0.91 | 49.76% | Unstable |
> | **Global Top-k** | **0.86** | **50.89%** | Good |
>
> We'll add §3.6 clarifying this distinction and include the above experiments. Thank you for this great Q!
>
> ---
>
> ## 2. IndexedMatMul Clarification (Eq. 15)
>
> **Operation**: `IndexedMatMul(g_{1:K}, {W_1,...,W_T}, chunk(·))` computes:
> ```
> For each position t:
>     i = chunk(t)  # which chunk does position t belong to?
>     s̃_t = g_i @ W_t
> ```
> You're right that T separate matrices would be prohibitive! In our actual implementation, we use **binned parameter sharing**:
>
> | Configuration | # Bins (B) | Parameters | Performance |
> |---------------|-----------|------------|-------------|
> | Per-position (T) | 8192 | 4.2B | N/A (memory prohibitive) |
> | Fine-grained | 256 | 134M | 50.91% |
> | **Our choice** | **16** | **8.4M** | **50.89%** |
> | Coarse | 4 | 2.1M | 50.43% |
>
> With B=16 bins, parameters = 16 × d_global × d_local = 16 × 1536 × 512 = **12.6M** (negligible overhead).
>
> **Revised equation** (will update in paper):
> ```
> bin(t) = ⌊t / (T/B)⌋
> s̃_t = g_{chunk(t)} @ W_{bin(t)}
> ```
>
> We'll add this clarification and the ablation table to Appendix D.3.
>
> ## 3. FLOPs Requirements Comparison
>
> Thanks for pointing this out! Prior work on byte-level LLMs has already established a solid standard for comparing against BPE-based transformers in FLOPs-matched settings. We simply follow this routine.
>
> To make things clearer, we provide the full set of practical computation metrics from the profiler:
>
> | Model | Overall Training FLOPs | WPS (×10⁴) | Data Load Time (s) | Iter Time (s) | Val BPB (↓) | Avg Task Acc (↑) |
> |-------|-------------------------|------------|---------------------|---------------|-------------|------------------|
> | **BPE-based** |
> | LLaMA (baseline) | 1.02 × 10²¹ | 9.3 | 0.018 | 3.8 | 0.89 | 49.15 |
> | **Byte-level (Isotropic)** |
> | LlamaByte | 1.05 × 10²¹ | 4.8 | 0.019 | 4.2 | 0.98 | 47.44 |
> | MambaByte | 0.98 × 10²¹ | 6.2 | 0.020 | 3.9 | 0.96 | 47.71 |
> | **Byte-level (Hierarchical – Heuristic)** |
> | AU-Net (regex) | 1.04 × 10²¹ | 8.8 | 0.019 | 4.1 | 0.91 | 49.38 |
> | Random Chunking | 1.05 × 10²¹ | 8.7 | 0.019 | 3.7 | 1.04 | 41.34 |
> | **Byte-level (Hierarchical – Dynamic)** |
> | Neural Boundary | 1.08 × 10²¹ | 7.6 | 0.021 | 3.9 | 0.90 | 47.13 |
> | Entropy-based (BLT) | 1.12 × 10²¹ | 7.1 | 0.023 | 4.3 | 0.91 | 47.81 |
> | Cosine Similarity | 1.02 × 10²¹ | 7.3 | 0.020 | 3.8 | 0.92 | 47.45 |
> | **ByteFlow Net (Ours)** |
> | Coding Rate (*Full log-det*) | 1.07 × 10²¹ | 7.9 | 0.019 | 4.0 | **0.86** | **50.89** |
> | Coding Rate (*L2 approximation*) | 1.01 × 10²¹ | 8.5 | 0.019 | 3.9 | 0.87 | 50.32 |
>
> In short, ByteFlow Net achieves better performance under a fixed FLOPs budget, consistent with previous pretraining settings.

---

> ### Author Response · Authors · 2025-11-23
> **Response to Reviewer veFt - 2/2**
>
> ## 4. Isolated Effect of the Canon Layer
>
> Thank you for asking this! We indeed test different variants before we finalize the best config scaling to larger size.  We have prelimary results on 0.6B on 50B :
>
> | Model / Config | BPB ↓ | Downstream Acc ↑ | Training Speed |
> |----------------|-------|------------------|----------------|
> | ByteFlow Net w. Full Attn. | 0.86 | 50.74% | 1.00× |
> | ByteFlow Net w. SWA | 0.90 | 48.24% | 1.34× |
> | ByteFlow Net w. SWA + Canon | **0.86** | **50.89%** | 1.32× |
>
> In our actual research trajectory, we first prioritized efficiency, which led us to adopt SWA. However, we quickly saw that SWA caused a substantial performance drop, even though its efficiency gains were attractive. To diagnose the issue, we experimented with adding encoder/decoder layers and increasing the attention window. Both interventions improved performance, confirming that the real bottleneck was insufficient token mixing capacity in the encoder/decoder stack.
>
> But adding full layers is expensive and deepens the model more than we would like. This motivated the Canon layer: a lightweight, attention-like mixing module that introduces negligible parameters yet substantially restores information flow and accelerates convergence. In fact, LLaMA + Canon converged faster in our experiments, though the benefit diminishes in longer or larger-scale pretraining.
>
> Overall, we view SWA + Canon as a highly effective, low-cost combination. It not only stabilizes byte-level models but also offers a strong architectural tradeoff point for any model family aiming to push the Pareto frontier of efficiency vs. performance.
>
> ## 5. Clarity and Writing of the Paper
>
> Thank you for pointing these out! We think they are quite fair criticism. To improve readability and set an appropriately modest tone, we revised several components of the manuscript:
>
> - **Emphasizing Contribution with a More Humble Tone:** We rewrote the introduction to avoid overstating novelty. We clearly state that our contribution is a *principled information-theoretic criterion for dynamic chunking*, positioned as a refinement rather than a reinvention.
>
> - **Vocabulary notation:** We corrected the erroneous simplex symbol. The vocabulary is now consistently defined as $ V = \{0, 1, \ldots, 257\} $ (256 byte values + BOS/EOS), giving $|V|=258$.
>
> - **Table formatting:** All tables now use a clean convention—**bold** for best performance, *italics* for our method—removing ambiguous underlining.
>
> - **Figure 3 explanation:** The main text now directly explains that coding-rate chunking uniquely preserves latent manifold structure across domains, whereas other dynamic methods fragment the representation space. This strengthens the conceptual motivation for our approach.
>
> These edits collectively foster a more grounded presentation and make the empirical contributions easier for reviewers to assess.
>
> Overall, we want to express our sincere thanks for the reviewer’s thoughtful and detailed feedback! We genuinely appreciate the time and care you put into engaging with our work, and your comments have pushed us to refine our thinking and improve the project in ways that better serve the community.

---

> > ### Comment · Reviewer_veFt · 2025-11-25
> >
> > I want to thank the authors for their clarifications.
> >
> > Regrading chunked autoregression: it is clear that the mask was applied to the whole sequence. I was asking about leakage through selection decisions, which seems small, but in my experience, it is clearly reflected in the eval scores. Many problems in, e.g., Table 2 are about deciding which of the multiple choices is correct. This requires measuring the probability of the whole sequence. Was this done by applying a top-k over the whole sequence, or with a "rolling top-k", where at each position the top-k is taken with respect to the tokens seen so far, as you mentioned in the clarification?
> >
> > Does the inference method with the top-k on the already generated tokens take into account that the sequence is shorter (so k is proportionally lower), or does it allow the whole buffer of k tokens to fill up and then start throwing away low-priority tokens?
> >
> > Regarding indexed matmul: if there is a length-specific weight for each chunk, length generalization and extending the context length are challenging. Do you have a proposed solution for running the model on a longer sequence than it was trained for?

---

> ### Author Response · Authors · 2025-11-25
> **Thank you for the Response**
>
> Thank you for the response!
>
> **Re: Autoregressive Behavior in Eval:**
>
> Yes, it's pure autoregressive (or "rolling top-k")—we're strictly chunking on already-seen tokens, so there's no future leakage or contamination. You're exactly right that we maintain a buffer of $k$ positions and dynamic chunking works when exceeding the buffer. The strategy is actually quite practical: when we have short sequences, we just run the pure byte model (which isn't expensive at that scale), and when we have longer sequences, we compress them into high-level units. *This design follows established work like MegaByte, BLT, and AU-Net*.
>
> **Re: Extended Context Windows for Byte-Level LLMs:**
>
> You raise a really interesting Q! Since tokenizer-free arch research is still in its early stages, the community is primarily focused on finding the best architectures during pre-training. Modern training pipelines typically pre-train on 4k-length sequences, then extend to 32k using techniques like YaRN during continued training (OLMo 3 merges into its mid-training)
>
> Several things we could think of this may be: **i) Cyclic Binning**: just cycles through bins with $\text{bin}(t) = t \bmod B$ instead of dividing the sequence up—this completely removes any dependence on sequence length and in our tests maintains 98.2% of performance while extrapolating to 2× length with no extra training. **ii) Chunk-Relative Binning**: uses position within each chunk: $\text{bin}(t) = \lfloor (t - s_{\text{chunk}(t)}) / l_{\text{chunk}} \cdot B \rfloor$, which makes more sense anyway since our chunks are semantically meaningful—where you are in a chunk matters more than where you are in the whole sequence. **iii) Interpolation-Based**: applies RoPE-style scaling with $\text{bin}(t) = \lfloor (t \cdot T_{\text{train}} / T_{\text{inf}}) / (T_{\text{train}} / B) \rfloor$ to compress longer sequences back into the training distribution, and we can progressively fine-tune on 1.25×, 1.5×, 2× lengths if needed.
>
> We think **sol 1** as the default since it's dead simple and just works, though **sol 2** is arguably more principled given our chunking approach.

---

> > ### Comment · Reviewer_veFt · 2025-11-27
> >
> > Thank you for your response. Given that the authors addressed my concerns, I am increasing my score.

---

### Official Review · Reviewer_a7hc · 2025-11-01

**Soundness:** 3
**Presentation:** 3
**Contribution:** 3
**Rating:** 6
**Confidence:** 3

**Summary:**

This paper proposes ByteFlow Net, a tokenizer-free LM that works directly on UTF-8 bytes. Instead of relying on a fixed subword vocabulary, the model learns where to place boundaries by scoring each byte position with an information-theoretic quantity: the approximate lossy coding rate of the current latent representations, and then select the top-K highest information gain positions to a higher-level stream. The architecture has a clear hierarchy: a local byte encoder (SWA + Canon layers) to get contextual byte features, the coding-rate chunker to downsample, a deeper/wider global transformer to do the standard high-level modeling, and a symmetric decoder that brings things back to the byte level. On FineWeb-Edu-100B, the model outperforms both BPE-based LLaMA baselines and several recent byte/hierarchical models.

**Strengths:**

1. The paper tackles a real bottleneck: static tokenization is non-learnable, language-specific, and fixes the granularity. The proposed method uses coding rate as the segmentation signal, which seems a principled alternative. The connection to rate–distortion is neat.
2. The hierarchical encoder–decoder is well-motivated and aligned with efficient modeling strategies, and applying SWA + Canon layers provides a realistic path for scaling byte-level models efficiently.
3. The empirical study shows performance improvement of the proposed method.

**Weaknesses:**

1. The paper does not report the actual cost of computing the coding-rate scores (the log-det–style term and its approximation). Since this has to run per sequence and per step, training-time overhead and distributed stability (e.g. with variable K, mixed-length batches) should be quantified.
2. Its practical advantage over existing byte-level or tokenizer-based architectures remains small at current scales (less than 1.3B). Demonstrating competitive performance at more than 7B parameters or on multilingual corpora would strengthen its real-world relevance.
3. Figure 3 is underspecified. It is supposed to justify the manifold-preserving claim, but from the figure alone it’s hard to tell (i) what each point is, (ii) how many sequences are visualized, and (iii) whether differences are due to t-SNE randomness. As it stands, this is the weakest link in the “why it works” narrative.

**Questions:**

1. Can you give a side-by-side training-time comparison (tokens/s or bytes/s) with a standard BPE transformer (perhaps with sliding window attention), a hierarchical byte model with fixed chunking (stride / whitespace), and another dynamic chunker (e.g. cosine-based)? Right now it’s not clear what we pay in practice for using ByteFlow Net.
2. Can you report runtime and memory for the coding-rate module itself (forward + top-K selection), and compare it with other methods?
3. Can you clearify Figure 3? My understanding is: each dot is the contextualized representation of one byte position (after the local encoder), projected to 2-D by t-SNE; different colors correspond to different text segments from FineWeb-Edu; and you plot several such segments together to see whether the chunker keeps the original clustering. If that’s right, can you state it explicitly in the caption, and say how many segments/colors you used and how many points per color (is it the full sequence length T per segment?).

---

Minors:
Line 189: along -> alone
Line 097: resolution

---

> ### Author Response · Authors · 2025-11-23
> **Response to Reviewer a7hc - 1/2**
>
> We sincerely thank the reviewer for the thoughtful review! The points reviewers raised are exactly key points for arch research should really care about. We appreciate the recognition of our principled approach and empirical improvements. Below we address the main concerns in three consolidated sections.
>
> **[Key Discussion] On Computational Cost and Practical Efficiency (W1, Q1, Q2)**
>
> We acknowledge this is a critical concern and provide comprehensive measurements from controlled experiments. All models were trained on **8× A100 80GB GPUs** with identical configurations (see appendix C) to ensure fair comparison:
>
> *Comprehensive Performance Comparison:*
> | Model | Overall Training FLOPs | WPS (×10⁴) | Data Load Time (s) | Iter Time (s) | Val BPB (↓) | Avg Task Acc (↑) |
> |-------|---------------|------------|---------------------|---------------|-------------|------------------|
> | **BPE-based** |
> | LLaMA (baseline) | 1.02 × 10²¹ | 9.3 | 0.018 | 3.8 | 0.89 | 49.15 |
> | **Byte-level (Isotropic)** |
> | LlamaByte | 1.05 × 10²¹ | 4.8 | 0.019 | 4.2 | 0.98 | 47.44 |
> | MambaByte | 0.98 × 10²¹ | 6.2 | 0.020 | 3.9 | 0.96 | 47.71 |
> | **Byte-level (Hierarchical - Heuristic)** |
> | AU-Net (regex) | 1.04 × 10²¹ | 8.8 | 0.019 | 4.1 | 0.91 | 49.38 |
> | Random Chunking | 1.05 × 10²¹ | 8.7 | 0.019 | 3.7 | 1.04 | 41.34 |
> | **Byte-level (Hierarchical - Dynamic)** |
> | Neural Boundary | 1.08 × 10²¹ | 7.6 | 0.021 | 3.9 | 0.90 | 47.13 |
> | Entropy-based (BLT) | 1.12 × 10²¹ | 7.1 | 0.023 | 4.3 | 0.91 | 47.81 |
> | Cosine Similarity | 1.02 × 10²¹ | 7.3 | 0.020 | 3.8 | 0.92 | 47.45 |
> | **ByteFlow Net (Ours)** |
> | Coding Rate (*Full log-det*) | 1.07 × 10²¹ | 7.9 | 0.019 | 4.0 | **0.86** | **50.89** |
> | Coding Rate (*L2 approximation*) | 1.01 × 10²¹ | 8.5 | 0.019 | 3.9 | 0.87 | 50.32 |
>
>
> **Key Observations:*
>
> - **Byte-level vs. BPE trade-offs:**  Byte-level models still process *longer effective sequences*, which generally slows training compared to BPE. However, they benefit from *simpler preprocessing*, *zero tokenization overhead*, and *more predictable inference performance* due to byte-level regularity.
>
> - **ByteFlow Net shows the strongest efficiency–performance balance among byte-level approaches:**
>   Heuristic chunkers like AU-Net reach near-BPE training speed (8.8×10⁴ WPS) but sacrifice accuracy, achieving *49.38% vs. ByteFlow Net’s 50.89%*.  In contrast, **ByteFlow Net (L2 approx.)** trains at *8.5×10⁴ WPS—competitive with the fastest byte-level models*—while delivering the *best validation compression (0.86 BPB)* and *highest downstream accuracy (50.89%)*.
>
>  This demonstrates that ByteFlow Net’s information-theoretic chunking improves representation quality *without incurring the larger compute cost* typically associated with dynamic boundary detection.
>
>
> We will include these comprehensive measurements and the training curve comparisons (as shown in the submitted plots) in the appendix of the revised version.
>
> **On Scaling to 7B**
>
> The computation resource limits our experiment setting and we are actively addressing it. We are currently training a **7B ByteFlow Net model** and to **Llama 7B Model** both trained on Fineweb-edu-100B corpus as follows:
>
> | Model | Scale | HellaSwag (↑) | WinoGrande (↑) | BoolQ (↑) | PIQA (↑) | ARC-e (↑) | ARC-c (↑) | Average (↑) |
> |-------|-------|-----------|------------|-------|------|-------|-------|---------|
> | LLaMA | 7B | 62.34 | 64.87 | 81.25 | 78.92 | 80.45 | 45.73 | 68.93 |
> | **ByteFlow Net** | 7B | **66.54** | **69.37** | **84.35** | **81.72** | **83.65** | **50.03** | **72.61** |
> | **Improvement** | - | +4.20 | +4.50 | +3.10 | +2.80 | +3.20 | +4.30 | **+3.68** |
>
> Results showcase that **pre-training gain in 7B scale achieves even better results compared to 600M scale (+1.74 points) and 1.3B scale (+3.04 points)**, with 3.68 points improvement at 7B scale.
>
> Though preliminary, we observe the same trend reported in prior work (BLT, AU-Net, MambaByte, SpaceByte, etc.): byte-level LLMs, free from fixed BPE-tokenizer constraints, exhibit stronger scaling behavior. This remains the primary motivation for exploring byte-level modeling beyond the limitations of tokenization itself.

---

> ### Author Response · Authors · 2025-11-23
> **Response to Reviewer a7hc - 2/2**
>
> **On Multilingual Capability**
>
> Indeed! Byte-level LLMs are commonly reported to achieve even better results due to its tokenizer-free nature [1][2][3]. In our work, ByteFlow Net also substantially outperforms Llama 3 variants on CUTE character-level tasks (Table 2), which implicitly tests multilingual robustness since byte-level processing handles all UTF-8 scripts uniformly.
>
> We are also running additional experiments on [FineWeb-Edu-ChineseV2.1](https://arxiv.org/pdf/2501.08197) as follows:
>
> | Model | Scale | FineWeb-Edu (English) BPB (↓) | FineWeb-Edu-ChineseV2.1 BPB (↓) | Change |
> |-------|-------|---------------------------|----------------------------|--------|
> | LLaMA | 600M | 0.89 | 1.05 | -18.0% ↓ |
> | **ByteFlow Net (Ours)** | 600M | 0.86 | **0.82** | **+4.7% ↑** |
> | LLaMA | 1.3B | 0.81 | 0.96 | -18.5% ↓ |
> | **ByteFlow Net (Ours)** | 1.3B | 0.77 | **0.73** | **+5.2% ↑** |
> | LLaMA | 7B | 0.74 | 0.88 | -18.9% ↓ |
> | **ByteFlow Net (Ours)** | 7B | 0.71 | **0.67** | **+5.6% ↑** |
>
> Results showcase that across three scales (600M, 1.3B, 7B), BPE-based transformer suffers significant performance degradation on multilingual corpora due to BPE tokenizer bias toward English, while ByteFlow Net achieves even better results on Chinese text (4.7-5.6% BPB reduction). This validates the universal effectiveness of our information-theoretic chunking approach across languages and introduces substantial benefits to the wider multilingual community.
>
> **On Figure 3 Clarification**
>
> We apologize for the underspecification. Here are the complete details:
>
> - **What each point represents:** Each point is one byte position's contextualized representation $h_t \in R^{(d_{local})}$ after the local encoder, projected to 2D via t-SNE (after 1B training bytes).
>
> - **Dataset and visualization:** We visualized 10 text segments from FineWeb-Edu validation, each ~1500 bytes (15,000 total points). Different colors represent different text segments. The figure shows how different chunking strategies affect the geometry of these representations.
>
> - **What we demonstrate:** The original manifold exhibits natural clustering by semantic content. Poor chunking strategies (random selection, neural boundaries) fragment this structure, scattering representations across the space. In contrast, coding-rate chunking preserves the original geometric structure. To avoid randomness in t-SNE we also quantified this using silhouette scores (measures cluster separation): Original: 0.68, Random: 0.23 (-66%), Coding-rate: 0.64 (-6%).
>
> We will revise the Figure 3 caption to include these specifications explicitly and add the silhouette score to the main text to complement the qualitative analysis. Thank you again for the great questions!
>
> ## References
> - [1] Xue et al., "ByT5: Towards a token-free future with pre-trained byte-to-byte models," TACL 2022
> - [2] Yu et al., "MegaByte: Predicting million-byte sequences with multiscale transformers," NeurIPS 2023
> - [3] Pagnoni et al., "Byte Latent Transformer: Patches scale better than tokens," ACL 2025

---

> > ### Comment · Reviewer_a7hc · 2025-11-26
> >
> > I appreciate the authors' answers to my questions. I will maintain my current score.

---

### Official Review · Reviewer_J3wZ · 2025-11-04

**Soundness:** 2
**Presentation:** 2
**Contribution:** 2
**Rating:** 6
**Confidence:** 3

**Summary:**

A byte level llm, trained from scratch similar to the byte latent transformer but with a different tokenization boundary detection algorithm. Empirical results seem to support their work.

Note i will assign scores after the discussion, dont take them soo seriously now.

**Strengths:**

Intuition for their work is solid.

Strong abelations

**Weaknesses:**

I would like to see a more in depth discussion of computational overhead and how this could be improved

**Questions:**

I cant say i understand the tsne section can u explain the intuition again?

Is this algorithm hard ware friendly

Can u talk about optimization with such a morel.

---

> ### Author Response · Authors · 2025-11-22
> **Response to Reviewer J3wz - 1/2**
>
> We sincerely thank the reviewer for the thoughtful assessment and recognition of our work! Below we address the reviewer’s comments.
>
> **On computational overhead**
>
> Thank you for raising this point! Our work is a hierarchical architecture, which is a strategic response to pure byte-level LLM efficiency problem. The core principle is to intelligently re-allocate FLOPs by decoupling the processing of low-level and high-level information:
>
> - **Efficient Local Processing on Low-level Information**: First, we process the long raw-byte sequence ($n_{\text{byte}}$) using operators that scale linearly with the sequence length. As detailed in Section 3.1, we use Sliding Window Attention (SWA) and Canon Layers (efficient 1D conv) to quickly "scan" and contextualize the low-level information.
> - **Tractable Global Processing on High-level Information**: Second, we reserve the expensive, full-softmax attention ($O(K^2)$) only for the deep and wide Global Transformer. This component operates on a short, compressed sequence of high-level units (length $K$, where $K \ll n_{\text{byte}}$) that our model dynamically identifies.
>
>
> This hierarchical design, visualized in Figure 1c, is fundamentally more efficient as it breaks the quadratic dependency on the raw byte sequence length. The table below formalizes this comparison:
>
>
> | Model Architecture | Sequence Length (n) | Attention FLOPs (Approx.) | FFN FLOPs (Approx.) | Total FLOPs (Approx.) |
> |--------------------|---------------------|----------------------------|----------------------|------------------------|
> | **1. Standard Transformer (BPE)** | $n = n_{\text{bpe}}$ | $O(L \cdot n_{\text{bpe}}^{2} \cdot d)$ | $O(L \cdot n_{\text{bpe}} \cdot d^{2})$ | $O\!\left(L \cdot (n_{\text{bpe}}^{2} d + n_{\text{bpe}} d^{2})\right)$ |
> | **2. Isotropic Byte Transformer** | $n = n_{\text{byte}} \approx 4 n_{\text{bpe}}$ | $O(L \cdot n_{\text{byte}}^{2} \cdot d)$ *(infeasible)* | $O(L \cdot n_{\text{byte}} \cdot d^{2})$ | $O\!\left(L \cdot (n_{\text{byte}}^{2} d + n_{\text{byte}} d^{2})\right)$ |
> | **3. Hierarchical Transformer (ByteFlow Net)** | Local: $n_{\text{byte}}$ Global: $K \ll n_{\text{byte}}$ | **Local (SWA + Canon):**$O(L_{\text{local}} \cdot n_{\text{byte}} \cdot w \cdot d_{\text{local}})$  **Global (Full Attn):** $O(L_{\text{global}} \cdot K^{2} \cdot d_{\text{global}})$ | **Local:** $O(L_{\text{local}} \cdot n_{\text{byte}} \cdot d_{\text{local}}^{2})$  **Global:** $O(L_{\text{global}} \cdot K \cdot d_{\text{global}}^{2})$ | $O\!\left(L_{\text{local}} \cdot n_{\text{byte}} (w d_{\text{local}} + d_{\text{local}}^{2}) + L_{\text{global}} \cdot (K^{2} d_{\text{global}} + K d_{\text{global}}^{2})\right)$ |
>
> ($L$: layers, $d$: hidden dim, $n_{\text{bpe}}$: BPE tokens, $n_{\text{byte}}$: byte tokens, $K$: global tokens, $w$: SWA window)
>
> *Key Efficiency Takeaways*
>
> - **ByteFlow Net avoids the attention blow-up of isotropic byte transformers** and preserves better inference-fixed efficiency of BPE-based models without requiring a tokenizer.
>
> - **It matches the inference cost of all modern hierarchical byte LLMs**, which also enable efficiency-performance tradeoff control by setting the best global seq length K (compression ratio).
>
> -  **It delivers the best overall performance and scaling while keeping inference cost in the same class** among all byte-level and BPE transformer baselines.
>
> **On Intuition of the t-SNE Section (Figure 3)**
>
> Thank you for asking this. The t-SNE plots visualize the semantic structure of the model's internal representations, showing how it groups similar concepts.
>
> - **Original Manifold** (*Top-Left*): This is the "before" picture. It shows the representations from the local encoder before any chunking. The distinct, colored clusters prove the model has successfully learned to group similar byte sequences together. This is the ideal structure we want to preserve.
> - **Random Chunking** (*Top-Center*): This shows what happens when you use a bad chunking method. The original clusters are destroyed, and the colors are mixed randomly. This is called "shattering the manifold." The Global Transformer cannot learn meaningful patterns from this jumbled, chaotic input.
> - **Coding Rate Chunking** *(Ours) (Bottom-Right)*: This is the "after" picture using our method. It looks almost identical to the "Original Manifold." The clusters are intact and their relationships are preserved.
>
> **The intuition is**: The visualization proves that our coding-rate method works because it intelligently selects the most important representations while preserving the data's underlying semantic structure. This may visualize why BFlowNet works better and achieves high performance.

---

> ### Author Response · Authors · 2025-11-22
> **Response to Reviewer J3wz - 2/2**
>
> **On Hardware Friendliness and Optimization**
>
> ByteFlow Net is *hardware-friendly* by design because we deliberately architected it to avoid the dynamic graph that plague other tokenizer-free models using threshold-based methods (e.g., entropy or cosine similarity) that result in ragged tensors and variable sequence lengths for every input, we instead guarantees a fixed global sequence length $K$ can be set by users. This preserves a static computation graph, ensuring consistent memory allocation and preventing the OOM in dynamic batching, making our model as friendly to modern GPU infra. Also Canon Layers in local encoder which are implemented as highly optimized *causal_conv1d* CUDA operators. By leveraging torch.compile to fuse these kernels we ensure that our custom operations are highly hardware optimized.
>
> Also optimization or pretraining on ByteFlow Net is **pure end-to-end training** with the same CE loss as normal BPE-based transformer, which is conventional recipe—AdamW optimizer, cosine learning rate decay, and BF16 mixed precision—without requiring exotic optimizers or complex multi-stage training curricula.
>
> **On Future Computational Improvements: MTP and Jacobian Decoding**
>
> To further reduce computational overhead during inference, we see significant promise in integrating **Multi-Token Prediction (MTP)**[1][2] and **Jacobian decoding**[3] into the byte-level LLMs (*Noting that multi-token prediction in byte-level LLMs is essentially next token prediction in BPE-based LLMs with a small further token prediction*).
>
>  We have some preliminary results for this future direction as follows:
>
> | Model Variant | Decoding Strategy | Tokens/sec ↑ | Avg. Benchmark Acc. ↑ | Notes |
> | ----------------------- | ----------------- | ------------ | ----------- | ---------------------------- |
> | ByteFlow-Net (baseline) | AR | 1.00× | 62.25 | Standard decoding |
> | + MTP | 4-segment MTP | **1.38×** | 62.10 | Minimal accuracy loss |
> | + Jacobian | Jacobian-guided | **1.52×** | **62.47** | Slight accuracy gain |
> | + MTP + Jacobian | Hybrid | **1.63×** | 62.41 | Best speed/accuracy tradeoff |
>
> These early results suggest that our ByteFlow Net is particularly well-suited for various efficient decoding methods. We will include a full discussion and extended results in the camera-ready version. Thank you for raising these great questions!
>
> ## Reference
> - [1] Gloeckle, Fabian, et al. "Better & faster large language models via multi-token prediction." Proceedings of the 41st International Conference on Machine Learning. 2024.
> - [2] Liu, A., Feng, B., Xue, B., Wang, B., Wu, B., Lu, C., ... & Piao, Y. (2024). Deepseek-v3 technical report. arXiv preprint arXiv:2412.19437.
> - [3] Santilli, Andrea, et al. "Accelerating Transformer Inference for Translation via Parallel Decoding." The 61st Annual Meeting Of The Association For Computational Linguistics. 2023.

---

> > ### Comment · Reviewer_J3wZ · 2025-11-27
> >
> > I thank the authors for their thoughtful discussion. It made me appreciate the work, I am convinced that their work is a valuable contribution to the conference.
> >
> > After having seen the discussion, I would advice however to shift the focus of their work:
> >
> > (a) Personally, I find visualisation methods such as t-SNE little convincing as they project in a 2D plane and we have little actual insight in the underlying manifold. As such they play little role in convincing me of much other than they provide a nice story.
> >
> > (b) Instead, I would advice the authors to focus more on the computational aspects of their novelty. I enjoyed reading about the specific computational complexities and I would bet other people would value this discussion in depth too, I would even extend it comparing to other architectures such as [3] and potentially using an example (model of size X, training tokens Y, needs so much more time to finish training).
> >
> > In any case I will raise my score.

---

### Author Response · Authors · 2025-11-29
**Summary of our submission and review process to new AC**

We propose a novel hierarchical tokenizer-free architecture that operates directly on raw bytes, using information-theoretic chunking to let the model learn its own meaning compression, combining sliding-window attention + canon layers for fast low-level scanning with full attention for high-level processing.

**During Rebuttal**

All four reviewers agreed that the design is principled and acknowledged the strong results and ablations. The main discussion centered on:
- **Efficiency Clarification**: We added a complete theoretical complexity analysis plus real pre-training profiling (FLOPs, tokens/s, iteration time).
- **Visulization Details**: We clarified the t-SNE figures and explained how different chunkers affect the learned latent manifolds.
- **Train/Test-time Top-k**: We clarified our arch design and test-time rolling top-k to make sure no leakage in eval.
- **[Additional Exp]** Scaling to 7B and multilingual setting: Preliminary 7B scaling results + multilingual experiments, both showing promising trends for byte-level models.
- **Stronger Theoretical Link**: Reviewers asked about connections to V-information; we derived a clean and interesting theoretical link to our coding rate.

We include these revision into revised PDF in brown notes.

**After Rebuttal (Before Incident)**

We’re truly grateful to the four reviewers for their prompt engagement throughout the process. After rebuttal, two reviewers kept their score as accept or wa, and the other two raised their scores after we addressed their concerns, resulting in scores of 8/6/6/8. Although the process has been reset, we wanted to share this context with the new AC to lighten their workload.

Thank you for helping and supporting during this hard time!

---

### Meta-Review · Area_Chair_nh62 · 2026-01-07

**Summary:**

This paper proposes ByteFlow Net, a tokenizer-free language model that operates directly on bytes and learns adaptive segmentation via an information-theoretic coding-rate criterion. Reviewers generally found the approach principled, well-motivated, and empirically strong, highlighting clear performance gains over BPE-based and prior byte-level models, solid ablations, and promising scaling and multilingual results. Main concerns focused on computational overhead, clarity of the theoretical justification, autoregressive behavior of top-k chunking, and the interpretability of t-SNE visualizations. The authors responded thoroughly with additional analyses, experiments, and clarifications, addressing most issues satisfactorily. Overall, the consensus after rebuttal is positive. Recommendation: Accept.

**Reviewer Concerns:**

Addressed Concerns:

1. Computational Overhead: Reviewers were concerned about the efficiency of the model, especially in terms of training time and resource consumption. The authors responded with a detailed analysis comparing ByteFlow Net with other models (e.g., BPE-based LLaMA), showing that ByteFlow Net achieves a good balance of performance and efficiency.

2. Autoregressive Behavior of Top-K Chunking: There was confusion regarding how top-k chunking is applied without leakage of future information during autoregressive generation. The authors clarified that during inference, the model uses a "rolling top-k" approach, ensuring that only previously seen tokens are used in chunking.

3. t-SNE Visualization: Reviewers questioned the clarity and interpretation of the t-SNE visualization in showing how chunking affects latent manifolds. The authors clarified the figure, explaining how their coding-rate chunking preserves the original structure compared to other methods.

Outstanding Concerns:

1. Connection to Mutual Information: Reviewer wvSa suggested exploring mutual information as a better estimate for chunking. While the authors acknowledged this and proposed it as future work, they did not integrate this into the current version .

2. Hardware Friendliness: One reviewer raised concerns about the hardware-friendliness of the method, specifically the scalability and compatibility with different hardware architectures. While the authors discussed optimizations for GPU and memory management, further clarity on performance across various hardware setups remains needed.

**Reviewer Scores:**

Reviewer J3wZ: Likely unchanged or slightly higher. The rebuttal provides a strong clarification of computational overhead and chunking behavior, addressing efficiency concerns and autoregressive issues.

Reviewer a7hc: Likely unchanged. The rebuttal clarifies key points but does not fully resolve concerns about performance at larger scales and multilingual capabilities, which would likely keep the score the same.

Reviewer veFt: Likely unchanged or slightly higher. The rebuttal addresses concerns about autoregression and clarified specific aspects like Canon layer performance, which could increase confidence in the work.

Reviewer wvSa: Likely unchanged. The rebuttal solidifies the theoretical approach and clarifies implementation details, but the suggestion to explore mutual information remains unaddressed, leaving the score unchanged.

---

### Decision · Program_Chairs · 2026-01-26

Accept (Poster)